# Characterization of Long-Time Series Variation of Glacial Lakes in Southwestern Tibet: A Case Study in the Nyalam County

**Ge Qu** [1], **Xiaoai Dai** [1,*], **Junying Cheng** [1], **Weile Li** [2], **Meilian Wang** [3], **Wenxin Liu** [1], **Zhichong Yang** [1], **Yunfeng Shan** [1], **Jiashun Ren** [1], **Heng Lu** [4,5], **Youlin Wang** [6], **Binyang Zeng** [7] **and Murat Atasoy** [8]

1 School of Earth Science, Chengdu University of Technology, Chengdu 610059, China
2 State Key Laboratory of Geohazard Prevention and Geoenvironment Protection, Chengdu University of Technology, Chengdu 610059, China
3 Department of Land Surveying and Geo-Informatics, The Hong Kong Polytechnic University, Hong Kong 999077, China
4 State Key Laboratory of Hydraulics and Mountain River Engineering, Sichuan University, Chengdu 610065, China
5 College of Hydraulic and Hydroelectric Engineering, Sichuan University, Chengdu 610065, China
6 Northwest Engineering Corporation Limited, Xi'an 710065, China
7 Southwest Branch of China Petroleum Engineering Construction Co., Ltd., Chengdu 610095, China
8 Department Geosciences, College of Sciences and Mathematics, Auburn University, Auburn, AL 36849, USA
* Correspondence: daixiaoa@mail.cdut.edu.cn; Tel.: +86-180-8198-8616

**Abstract:** Glacial lakes are important freshwater resources in southern Tibet. However, glacial lake outburst floods have significantly jeopardized the safety of local residents. To better understand the changes in glacial lakes in response to climate change, it is necessary to conduct a long-term evaluation on the areal dynamics of glacial lakes, assisted with local observations. Here, we propose an innovative method of classification and stacking extraction to accurately delineate glacial lakes in southwestern Tibet from 1990 to 2020. Based on Landsat images and meteorological data, we used geographic detectors to detect correlation factors. Multiple regression models were used to analyze the driving factors of the changes in glacier lake area. We combined bathymetric data of the glacial lakes with the changes in climatic variables and utilized HEC-RAS to determine critical circumstances for glacial lake outbursts. The results show that the area of glacial lakes in Nyalam County increased from 27.95 km$^2$ in 1990 to 52.85 km$^2$ in 2020, and eight more glacial lakes were observed in the study area. The glacial lake area expanded by 89.09%, where we found significant growth from 2015 to 2020. The correlation analysis between the glacial lake area and climate change throughout the period shows that temperature and precipitation dominate the expansion of these lakes from 1990 to 2020. We also discover that the progressive increase in water volume of glacial lakes can be attributed to the constant rise in temperature and freeze–thaw of surrounding glaciers. Finally, the critical conditions for the glacial lake's outburst were predicted by using HEC-RAS combined with the changes in the water volume and climatic factors. It is concluded that GangxiCo endures a maximum water flow of $4.3 \times 10^8$ m$^3$, and the glacial lake is in a stable changing stage. This conclusion is consistent with the field investigation and can inform the prediction of glacial lake outbursts in southwestern Tibet in the future.

**Keywords:** glacial lakes; information extraction; remote sensing; Qinghai–Tibet Plateau; climate response

## 1. Introduction

Glacial lakes, which are formed by the convergence of meltwater, are essential freshwater resources of local residents [1]. They are also hot spots of natural disasters in the Qinghai–Tibet Plateau [2]. In recent years, the warming of the Qinghai–Tibet Plateau has gradually led to significant changes in the number and size of glacial lakes [3–6]. Most of the glaciers in southeastern Tibet began to shrink and melt in the mid-19th century [7].

Especially after 2000, the melting of glaciers and the loss of materials have intensified [8,9], resulting in the increased area and frequency of changes in glacial lakes [10,11]. The glaciers in this area account for about 81.6% of China's total reserves, and it is one of the areas with the largest ice avalanches in China [12]. These glaciers are the source of many rivers [13], such as PumQu, Boqu, Tamba Kosi, and Lapchekhun Khola. Still, they are also essential in glacial lake disasters, including many steep mountains and fault zones, which are more likely to cause ice avalanches [14]. The abnormal changes of glacial lakes inevitably lead to a significant number of natural disasters, such as the outbursts of glacial lakes [15,16]. Some scholars have found that glacial lakes continue to develop [17,18], and more in-depth research is needed referring to the impact of climate change on the changes and eruptions of glacial lakes [19].

To effectively map glacier lakes across a mountain region [20], optical remote sensing imagery shows obvious advancement in estimating ice phenology of lakes and rivers around the globe over in situ measurement methods [21]. Optical sensors such as MODIS (moderate-resolution imaging spectroradiometer) have relatively low resolution and relatively short observation periods. The spatial resolution of Landsat TM, ETM+, and OLI images can reach 30 m, which is more conducive to observing glacial lakes and long-term trend analysis [22]. Most imagery is susceptible to cloud cover due to the Indian Ocean monsoon. However, the collection of no-cloud remote sensing imagery is often susceptible to cloud-covered imagery, further affecting the accuracy of contouring [23]. In southwestern Tibet, most glacial lakes are heavily covered by fragments of rock fragments and surface moraine [24]. The surrounding area of the lakeshore is covered by debris, mainly composed of rock fragments and surface moraines [17,25]. These glacial lakes are difficult to distinguish from hillsides due to the spectral similarity between surface debris and their surroundings [26]. To improve the accuracy of extracting glacial lakes, conventional approaches focus on combining multisource data to enhance recognition features beyond spectral features [27], improving or proposing new classification algorithms (e.g., support vector machines, random forest, and k-nearest neighbor), and relying directly on manual interpretation for complex domains [28]. To improve the accuracy of extracting glacial lake information based on the water body index method and digital elevation model, we proposed a classification and stacking extraction method to extract glacial lake boundaries under different states. The results show that the proposed threshold combined with the classification overlay extraction method can extract glacial lake boundaries in three states: unfrozen, semifrozen, and fully frozen, and can reduce the influence of glacier and hillshade and improve the utilization of optical images. In addition, few studies link a glacial lake's water level with climate to simulate the outburst of glacial lakes [29]. The characteristics of glacial lakes illustrate significant differences with respect to spatial and temporal scales [30]. How climate change affects the hydrological changes in southwest Tibet requires a comprehensive examination, remaining challenging to the existing research [31]. Currently, the methods of correlation analysis of glacial lakes are mostly Pearson correlation coefficient [32], principal component analysis, geographically weighted regression analysis, etc. Geographic detectors are mainly used to analyze the driving forces and influencing factors of various phenomena and multifactor interactions, which can quantify the strength of the interaction among influencing factors. This paper innovatively uses geographic detectors to study the driving factors of glacial lake changes, combined with multiple linear regression models and the Mann–Kendall trend test and estimation of Sen's slopes to accurately measure the impact of climate change on glacial lake changes. Considering the potential impact of glacial lakes on the local population in the southwestern Tibetan Plateau [33], it is imperative to study the changes in glacial lakes over the past 30 years [34]. A glacial lake outburst flood (GLOF) is a typical outburst flood caused by the failure of a dam containing a glacial lake, which consists of glacial ice or a terminal moraine [35]. Several GLOF studies have found that glacial retreat was changing water flow patterns, affecting the incidence of glacial lake outburst floods and increasing the risk of flooding and water scarcity associated with future climate change [36].

However, to accurately predict the changes of glacial lakes, it is necessary to predict their distribution characteristics and the underlying laws to understand the dynamic response of climate change to hydrological processes [37,38] and to study the eruption disasters of glacial lakes [39]. In contrast to empirical models, physical models consider the interaction between fluid and particles and flow behaviors such as turbulence during the simulation for glacial lake outburst simulations. Currently, commonly used physical numerical simulation models, including FLDWAN [40], BASEMENT [41], HEC-RAS [42], FLO-2D [43], etc., have been applied to modeling and analyzing glacial lake outburst floods in different regions of the world. This paper uses the HEC-RAS V.5 model to simulate and analyze the outburst of glacial lakes by combining the observation of water volume change information and bathymetric data. At the same time, according to the influence of climatic factors on the glacial lake and the trend of the long sequence change of the glacial lake, the prediction and analysis of the hazard of glacial lake outburst are carried out.

Here, we investigate the spatial and temporal changes in the size and location of several glacial lakes in Nyalam County from 1990 to 2020 using Landsat images and analyze the spatial distribution and characteristics of different types of glacial lakes in the study area from the perspectives of elevation, watershed, and mountain system. Furthermore, the effect of elevation on the changes of the glacial lake was estimated by analyzing the digital elevation model (DEM) and the changing pattern of the glacial lakes. The impacts of climate change and the corresponding drivers are further assessed. The monitoring and evaluation of the outburst hazards of glacial lakes are finally carried out by combining water volume change and bathymetric, can help provide essential data support for scientific assessment and prevention of glacial lake disaster risks.

## 2. Study Area and Data

### 2.1. Study Area

The study area is located in the Qinghai–Tibet Plateau in southwestern China (Figure 1). This study selects the upper reaches of Poqu, Nyalam County, Shigatse Region, and Tibet Autonomous Region to understand further the various characteristics of glacial lakes in this region. The coordinate of the study area is $86°00'00''E \sim 86°15'00''E$ and $28°00'00''N \sim 28°20'00''N$. Nyalam County is located in the southwestern part of the Tibet Autonomous Region, near the edge of the Himalayan Mountains and close to the Lhari Gangri Range, with a 2100 km$^2$ surface area. In 2020, 119 glaciers and 366 glacial lakes were recorded in Nyalam County. These glaciers are mainly located in the northeast and south-central regions of the county, with 98.63 km$^2$ [44]. Glacial lakes are primarily concentrated in the northeastern part of Nyalam County, and the junction of the Yala township glacial lake distribution area is relatively large. Most of them are moraine lakes or glacier blockage lakes, such as PacuCo, Chama QudanCo, DareCo, etc.

The geomorphology of the study area presents an alternating pattern of alpine valleys and plateau lake basins. Among them, large glacial lakes are formed in a concentrated distribution near the alpine glaciers, and small glacial trough lakes and ice lakes are scattered and distributed in the valleys. The region is mainly a subarctic semiarid and humid subtropical climate, with a distinct rainy season. There is an excellent possibility that the glacial lake will burst due to continuous rainfall.

### 2.2. Data Source and Preprocessing

Landsat multispectral images from 1990 to 2020 were used to extract the number and area of glacial lakes. As the glacial lake morphology is relatively stable in autumn and winter, this paper selects the images from September to November each year. This is also the season with low cloud cover and lake ice cover. In addition, less cloudy distribution images were chosen to avoid cloudiness's influence on the extraction boundary. Other auxiliary data such as DEM, meteorological data, and zoning data were applied to study the distribution of the number and area of glacial lakes and the relationship with meteorological factors (Table 1).

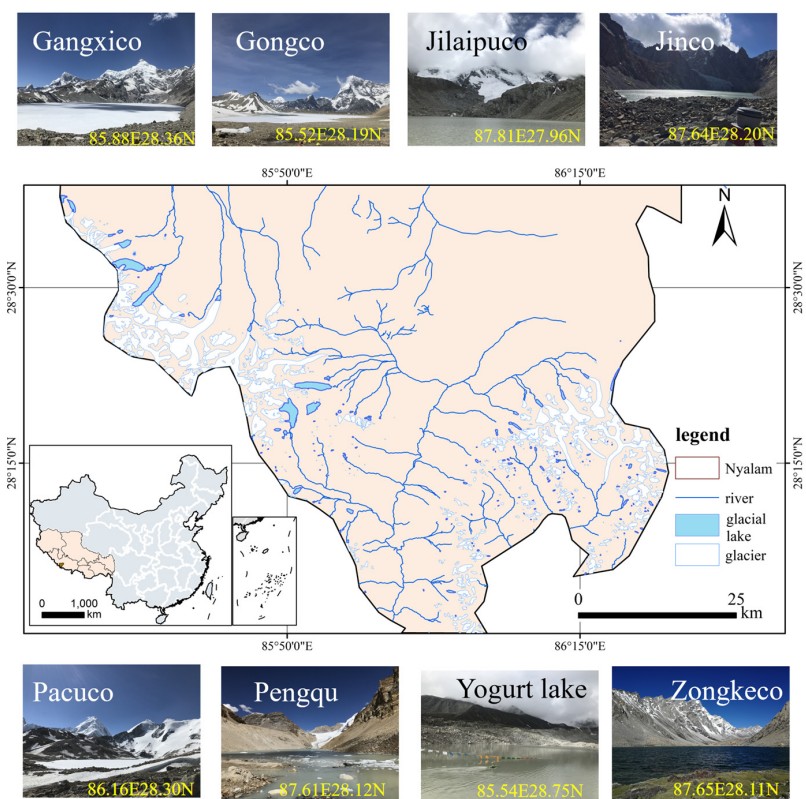

**Figure 1.** Study area and distribution of glacial lake area change.

**Table 1.** Data of the study area.

| Data Classification | Name | Resolution | Data Source |
|---|---|---|---|
| Remote sensing imagery data | Landsat4/5 TM | 1990.9.15 | USGS (https://glovis.usgs.gov/ (accessed on 1 January 2021)) |
| | Landsat4/5 TM | 1995.10.13 | |
| | Landsat7 ETM+ | 2000.11.4 | |
| | Landsat5 TM | 2005.11.12 | |
| | Landsat8 OLI | 2010.11.16 | |
| | Landsat8 OLI | 2015.11.24 | |
| | Landsat8 OLI | 2020.10.9 | |
| | GF-1 | 2018.10.10 | Geospatial Data Cloud (http://www.gscloud.cn/ (accessed on 1 January 2021)) |
| | GF-1 | 2019.10.5 | |
| | SRTM | DEM data | USGS (https://glovis.usgs.gov/ (accessed on 1 January 2021)) |
| Meteorological data | Meteorological station | Forms for report | Meteorological Data Center of China Meteorological Administration (http://cdc.cma.gov.cn/home.do (accessed on 1 January 2021)) |
| Zone | Administrative boundary vector | Vector data | Resource and Environment Science and Data Center (https://www.resdc.cn/ (accessed on 1 January 2021)) |

## 3. Methods

To investigate the spatial and temporal changes of glacial lakes in Nyalam County and their response to climate change, GangxiCo was selected as the study area. This paper used classification and superposition methods to extract glacial lake information automatically. Differences in area and distribution characteristics were sorted and analyzed for 30 years, from 1990 to 2020. The response mechanism of glacial lakes to other factors was then investigated. The multivariate linear simulation was then used to analyze the importance of climate change on the glacial lakes and reveal the changes in the glacial lake. The calculation of water level–flow of the glacial lake employed Manning's equation. The HEC-RAS was applied to determine the future trends of the glacial lake. The specific technical method proposed in this study was illustrated in the following subsections. Figure 2 shows the flow chart of our methodology.

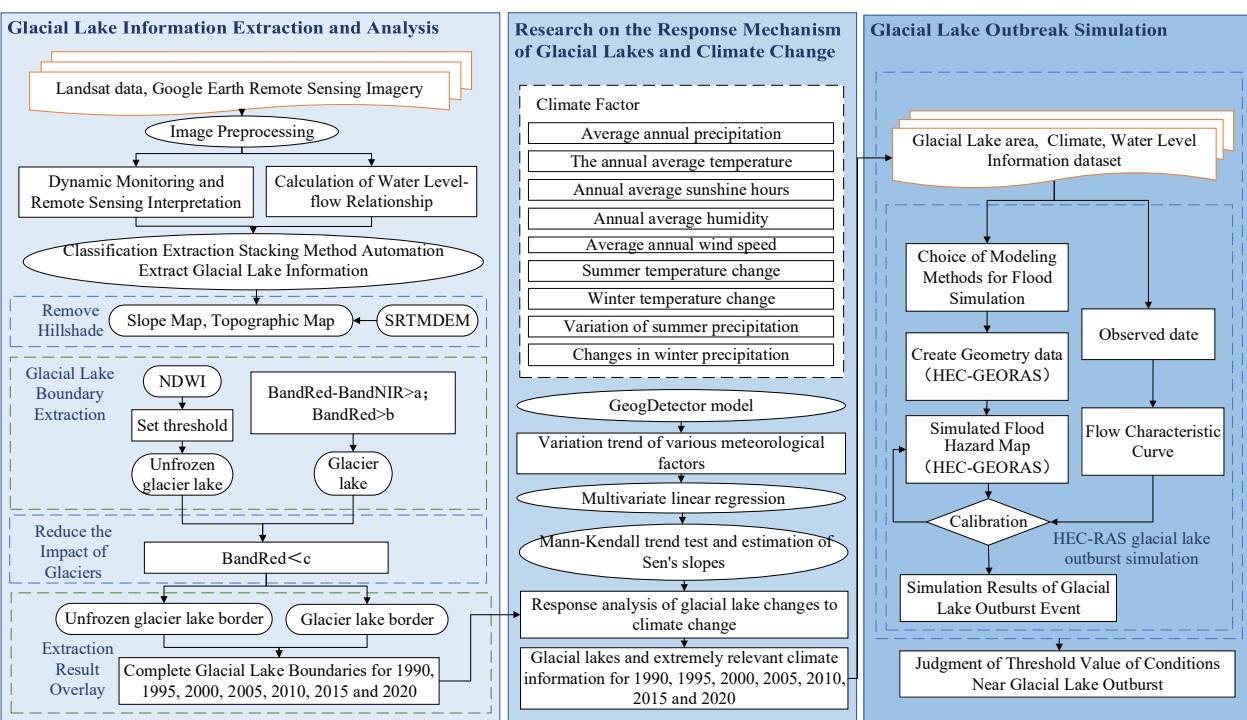

**Figure 2.** Flow chart of the methodology.

### 3.1. Glacial Lake Boundary Extraction

#### 3.1.1. Unfrozen Frozen Lake Boundary Extraction

This paper uses the calibration coefficients in the Landsat satellite header file. First, the radiometric calibration was performed using the software ENVI 5.1. Then, the surface reflectance values of each Landsat image were obtained by multispectral atmospheric correction conducted by the FLAASH model. Finally, this paper combines the normalized difference water index (NDWI) with the threshold method to extract the unfrozen glacial lake boundary. The NDWI uses the ratio of the green band to the NIR band to highlight the image's water body information and suppress vegetation's interference information to the maximum extent. Its calculation Equation (1) is as follows:

$$NDWI = (BandGreen - BandNIR)/(BandGreen + BandNIR) \qquad (1)$$

where BandGreen is the reflectance in the green band (Landsat TM, Landsat7 ETM+ band2, Landsat8 OLI band3) and BandNIR is the reflectance value in the NIR band (Landsat TM, Landsat7 ETM+ band4, Landsat8 OLI band5). The whole image was subjected to NDWI operation, and the brightness of different features was divided. Extraction of the unfrozen lake was implemented through a threshold filter. This paper establishes a significant threshold value (0.28) for extraction. In this way, all ground objects except the water of the unfrozen glacial lake were filtered as the background, leaving only the unfrozen glacial lake. Finally, the boundary of the unfrozen part of the glacial lake water body was obtained.

#### 3.1.2. Single-Band Threshold Boundary Extraction

After obtaining the water boundary of the glacial lake in the unfrozen part through boundary extraction, the boundary was obtained based on the spectral characteristics of the glacial lake. Because the difference between the reflectance of the glacial lake in the red band and the NIR band is more significant than a certain threshold, it was usually more remarkable than a specific value in the red band. Thus, Equations (2) and (3) were developed according to these two spectral characteristics of frozen glacial lakes. Considering that frozen glacial lakes have similar spectral characteristics to glaciers, this paper combined the regional attributes of Nyalam County. An experimental plan was set

up to magnify the reflectance 10,000 times, highlight it, and then calculate it [45]. Finally, the most suitable threshold for the study area was obtained according to the results extracted from the glacial lake and the Google Earth map. Through continuous analysis of the glacial lakes in Nyalam County, it was found that when a = 1800 and b = 9100, the boundaries of the extracted frozen glacial lakes are relatively complete. Most glacial lakes were formed by the convergence of meltwater produced by the retreat of glaciers and are distributed around the glaciers. The glacier represents strong reflection in the red band, and an appropriate threshold is set in the red light band to eliminate the influence of the glacier in the research area. Using Equation (4), we find that the impact of glaciers in the extraction process of glacial lakes can be effectively reduced when c is 12,000 through continuous analysis of glacial lakes in Nyalam County.

$$\text{BandRed} - \text{BandNIR} > a \tag{2}$$

$$\text{BandRed} > b \tag{3}$$

$$\text{BandRed} < c \tag{4}$$

where BandRed is the reflectance of the red band (Landsat TM, Landsat7 ETM+ band3, Landsat8 OLI band4) and BandNIR is the reflectance of the NIR band (Landsat TM, Landsat7 ETM+ band4, Landsat8 OLI band5). Since the surrounding environment of the glacial lake is generally relatively gentle, the topographic relief is slight, and the slope is usually tiny, nearly 0. The hillshade is attached to the backside of the mountain ridgelines, the topographic relief is significant, and the topographic slope varies greatly. This paper uses digital elevation model (DEM) data to distinguish lake and water pixels according to the above characteristics.

### 3.1.3. Precision Inspection

To further test the extraction method's performance and the new index's validity, the confusion matrix and overall classification accuracy were calculated, and the manual visual interpretation results of glacial lake boundary extraction were imported into Google Earth for rectification. The confusion matrix is constructed using ENVI 5.1 and calculates overall classification accuracy. The extraction accuracy of typical glacial lakes is evaluated by the confusion matrix's Kappa coefficient and the misclassification error and omission error. The Kappa coefficient Equation (5) is as follows:

$$K = \frac{P_0 - P_\varepsilon}{1 - P_\varepsilon} \tag{5}$$

where $P_0$ is the sum of the number of correctly classified samples in each category divided by the total number of samples. Suppose the number of actual samples in each category is $a_1, a_2, a_3, \ldots a_f$, the number of predicted samples in each category is $b_1, b_2, b_3, \ldots b_f$, and the total number of samples is n, then there are:

$$P_\varepsilon = \frac{a_1 \times b_1 + a_2 \times b_2 + \cdots a_f \times b_f}{n \times n} \tag{6}$$

### 3.2. Response Analysis of Glacial Lake Changes to Climate Change

#### 3.2.1. GeogDetector Model

The GeogDetector model quantifies the strength of various interactions between different ecological factors. The model follows an approach based on the assumption of spatial heterogeneity and is suitable for environmental research fields with high spatial heterogeneity. This paper uses the GeogDetector model to determine the spatial correlation between ecological factors and glacial lake changes. There are four detectors in the GeogDetector model, namely risk, factor, ecological, and interaction [46]. In this study, the differential factor and interaction detectors were used to quantitatively investigate the influence of

environmental factors on the changes in the glacial lake. In the process of divergence and factor detection, detecting the spatial variation of Y and detecting how much of the spatial divergence of attribute Y are explained by a specific factor, X. Using the q-value metric, the expression is:

$$q = 1 - \frac{\sum_{h=1}^{L} N_h \sigma_h^2}{N\sigma^2} = 1 - \frac{SSW}{SST} \tag{7}$$

$$SSW = \sum_{h=1}^{L} N_h \sigma_h^2 \, , \, SST = N\sigma^2 \tag{8}$$

where h = 1, ... , L is the stratum of variable Y or factor X, that is, the classification or partition; Nh and N are the number of cells in stratum h and the whole area, respectively; $\sigma_h^2$ and $\sigma^2$ are the variance of Y values in stratum h and the entire region, respectively. SSW represents the within sum of squares, and SST represents the total sum of squares. The value of q is in the range [0, 1], and the larger the value, the more pronounced the spatial differentiation of Y. If the independent variable X generates the stratification, then the larger value of q indicates the stronger explanatory power of the independent variable X on the attribute Y, and the weaker the opposite. Interaction detectors were then implemented to assess the combined effects of pairwise glacial lake changes.

### 3.2.2. Multivariate Linear Regression

This paper uses the main climate factors screened by the GeogDetector model analysis as independent variables. The multiple regression model is used to calculate the trend and annual change of the long-term series data of the glacial lake. The multiple regression model combines the relationship between the response and predictor variables. The general expression form of the multiple linear regression computational model is:

$$Y = a_0 + a_1x_1 + a_2x_2 + \cdots a_nx_n + C \tag{9}$$

where Y is the expected response value, $a_0$ is the model intercept, x is the set independent variable, n is the number of independent variables x, and C is the random error after removing the influence of n independent variables on Y.

### 3.2.3. Mann–Kendall Trend Test and Estimation of Sen's Slopes

This paper applied the Mann–Kendall test to find significant trends for each climate variable, with separate Mann–Kendall tests for each season or month. The Sen slope, which estimates the median rate of linear change over time, is also calculated to quantify the magnitude of the change. Finally, we verified the significance of lake area trends in southwestern Tibet. To examine the responses of trends to geographic and lake morphology predictors, we used multiple linear regression, using temperature, precipitation, humidity, insolation, and wind speed as model predictions and Sen's slope values as responses to identify trend patterns in glacial lake variables. In this study, the statistics Z-value, *p*-value, and Sen slope estimator slope value of the M-K test were calculated by R version 4.1.3.

### *3.3. Risk Analysis of Glacial Lake Outburst*
### 3.3.1. Water Flow Calculation

The research estimate is based on the field survey of GangxiCo glacial lake on 24 May 2019, to study the reflection of a glacial lake with climate and the glacial lake outburst and the water level–flow relationship of a glacial lake using Manning's equation to provide water body data for the later evaluation of glacial lake hazard. Moreover, joint glacial lake remote sensing data were used for outburst simulation. For the water level–flow estimates of natural rivers, the Manning Equations (10)–(12) was used as shown below:

$$Q = A\frac{1}{n}R^{\frac{2}{3}}S^{\frac{1}{2}} \tag{10}$$

$$R = \frac{A}{L} \tag{11}$$

$$Q = S^{\frac{1}{2}} \frac{1}{n} \frac{A^{\frac{5}{3}}}{L^{\frac{2}{3}}} \tag{12}$$

where A is the overwater cross-sectional area, n is the channel roughness, R is the hydraulic radius, S is the slope drop, Q is the flow quantity, and L is the wet perimeter.

The water level–flow equation derived from Manning's equation is the relationship equation of the downstream river channel of GangxiCo. Combined with remote sensing images and downstream longitudinal section measurement results, the estimated river channel slope drop is S = 0.048. The channel sediments are mainly coarse sand and gravel formed by water scouring after the weathering of granite gneiss [47], referring to the "Hydraulic Calculation Manual" (Second Edition). The roughness ratio of n = 0.04 is comprehensively selected according to the cross-sectional shape of the riverbed and the beach structure [48]. Finally, combined with the "Tibet Autonomous Region Shigatse Glacial Lake Surveying and Mapping Project," the water level–flow formula of the outlet section and the elevation obtained by the Manning formula is the relationship formula of the channel in the lower reaches of GangxiCo.

When the water surface elevation ≤5181.01 m, the water depth h ≤ 0.15 m, which can be shown as Equations (13)–(15):

$$A = 57.29 * h^2 \tag{13}$$

$$L = 114.60 * h \tag{14}$$

$$Q = 197.66 * h^3 \tag{15}$$

When the water surface elevation is between 5181.01 m~5185.51 m, that is, 0.15 < h < 4.65 m, it can be concluded that:

$$A = 1.29 + 0.5 * (17.19 * 2 + 20.94 * h) * h \tag{16}$$

$$L = 17.19 + 21.04 * h \tag{17}$$

$$Q = 5.48 \times \frac{\left(1.29 + 17.19h + 10.47h^2\right)^{\frac{5}{3}}}{(17.19 + 21.04 * h)^{\frac{2}{3}}} \tag{18}$$

### 3.3.2. HEC-RAS V.5 Model

The HEC-RAS is one of the most popular open-source models for glacier hazard studies. The glacial lake burst model of HEC-RAS combines a parametric burst model and a physical model. The basic principle is to use a parametric model to predict the parameters of the rupture and introduce a simplified physical dam rupture mechanism, which is used for the development of the crack to the final burst and to solve the process of the downstream propagation of the ruptured flood. The outburst process in this study means that when the flood volume increases to a specific value, it overflows the boundary of the glacial lake, scours the downstream to form an undercut, and gradually expands until the water flow continues to erode down both banks through the breach. This paper applies the new HEC-RAS V.5 model to solve the complete 2D diffuse wave equation given as:

$$\frac{\partial \delta}{\partial \tau} + \frac{\partial p}{\partial x} + \frac{\partial q}{\partial y} = 0 \tag{19}$$

$$\frac{\partial p}{\partial \tau} + \frac{\partial}{\partial x}\left(\frac{p^2}{h}\right) + \frac{\partial}{\partial y}\left(\frac{pq}{h}\right) = \frac{n^2 pq\sqrt{p^2 + q^2}}{h^2} - gh\frac{\partial \delta}{\partial x} + pf + \frac{\partial}{\rho \partial x}(h\tau_{xx}) + \frac{\partial}{\rho \partial y}(h\tau_{xy}) \tag{20}$$

where $h$ is the water depth (m), $p$ and $q$ are the specific flow rates in the $x$ and $y$ directions (m/s), $\delta$ is the surface elevation (m), $g$ is the acceleration of gravity (m/s$^2$), $n$ is the Manning resistance, $\rho$ is the water density (kg/m$^3$), $\tau_{xx}$ and $\tau_{xy}$ are the practical shear stress

components, and $f$ is the Coriolis force (s$^{-1}$). The inertia term of the momentum equation when the diffusion wave is chosen.

## 4. Results

### 4.1. Glacial Lake Extraction Results

To verify the extraction accuracy of the glacial lakes, the glacial lakes in three states of unfrozen, semifrozen, and fully frozen in 2020 were extracted as typical glacial lakes for accuracy analysis. The proposed classification extraction superposition algorithm in this paper first needs to set the threshold value for NDWI results to remove unfrozen glacial lakes. Then, a threshold is set for the difference between the red and NIR bands. Next, the frozen lake is extracted, and the red band is used to reduce the influence of glaciers. Finally, the unfrozen lake boundary is superimposed with the frozen lake boundary to obtain the complete glacial lake boundary. Figure 3a–c are remote sensing images, and Figure 3a1–c1 are the glacial lake boundaries by superimposition. In the figure, a is the remote sensing image of the unfrozen glacial lake, a1 is the extraction result of the unfrozen glacial lake, b is the remote sensing image of the not-fully frozen glacial lake, b1 is the extraction result of the fully frozen glacial lake boundary, c is the remote sensing image of the semifrozen glacial lake, and c1 is the extraction result of the unfrozen part of the semifrozen glacial lake.

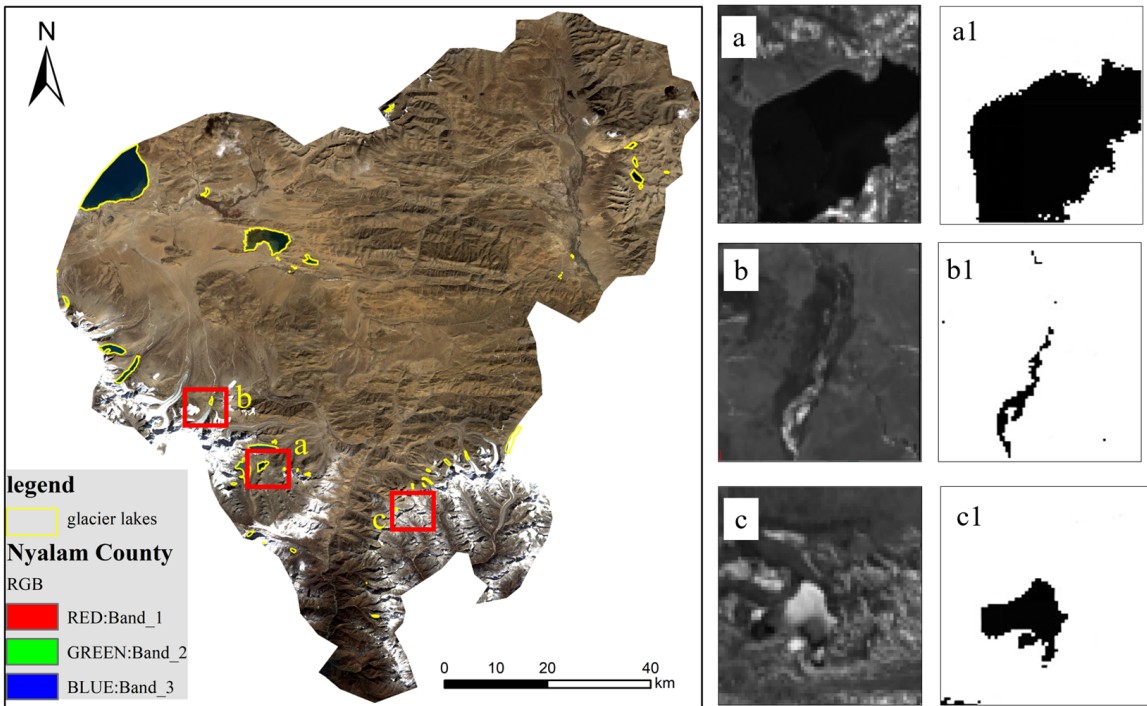

**Figure 3.** Remote sensing extraction map of the glacial lake area (16 November 2018).

4.1.1. General Distribution Characteristics of Glacial Lakes

1. Number and area changes

The above method obtained information on glacial lakes in 1990, 1995, 2000, 2005, 2010, 2015, and 2020 (Figure 4a). Statistical analysis was carried out on the research results, and the characteristics of changes in the number and area of glacial lakes from 1990 to 2020 were obtained (Table 2). It was found that during the 30 years from 1990 to 2020, the number of glacial lakes in Nyalam County increased by 7. The area increased by 24.90 km$^2$. Among them, the number growth rate was 0.12%, and the area growth rate was 89.12%. The data indicated that although the area of glacial lakes increased dramatically in the last three decades, the number of glacial lakes has not grown substantially simultaneously. Between 2000 and 2010, the number of lakes decreased, primarily due to the disappearance

of small glacial lakes near steep mountains and the merging of broken glacial lakes due to the increase in glacial meltwater. Therefore, while the number decreased, the area continued to increase [49]. After 2010, smaller glaciers were more sensitive to climate change because smaller glaciers reduced their area at a higher rate than more enormous glaciers [50]. According to the current situation, glaciers will further decrease, and the area of glacial lakes will continue to increase [51].

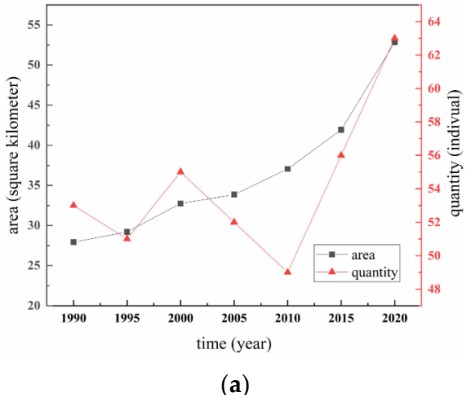

(**a**)

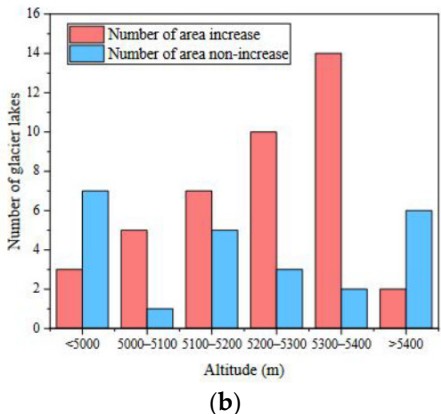

(**b**)

**Figure 4.** Changes in the number and area of glacial lakes from 1990 to 2020. (**a**) Changes in the number and area of glacial lakes from 1990 to 2020; (**b**) Changes in the number of area increases in glacial lakes from 1990 to 2020.

**Table 2.** Changes in the number and area of glacial lakes from 1990 to 2020.

| Time (Year) | Area (km$^2$) | Number (pcs) | Change in Area from the Previous Period (km$^2$) | Change in Number from the Previous Period (pcs) |
|---|---|---|---|---|
| 1990 | 27.942658 | 53 | | |
| 1995 | 29.210104 | 51 | 1.267446 | −2 |
| 2000 | 32.739735 | 55 | 3.529631 | 4 |
| 2005 | 33.861283 | 52 | 1.121548 | −3 |
| 2010 | 37.055987 | 49 | 3.194704 | −3 |
| 2015 | 41.941051 | 56 | 4.885064 | 7 |
| 2020 | 52.846172 | 63 | 10.90512 | 7 |

The area time series of six glacial lakes with rapid changes in the area between 1990 and 2020 (Figure 4b), shows continued lake growth. According to the overall analysis of glacial lake changes, the area of lakes in this region continues to grow. At the same time, the number of lakes has also grown to a lesser extent. From the overall situation of the changing trend of glacial lakes, the number and area of glacial lakes are increasing. Among them, the area of glacial lakes grew the fastest from 2015 to 2020, with a growth rate of 0.26 km$^2$·(5a)$^{-1}$. From 2000 to 2005, the area of glacial lakes grew the slowest, with a growth rate of 0.08 km$^2$·(5a)$^{-1}$. From 1995 to 2000, the area of glacial lakes increased moderately (More details in Supplementary Materials).

2. Altitude analysis

Historically, the Nyalam region has experienced four glacial periods, forming moraine landforms and lakes at different elevations. Therefore, the elevation distribution reflects the changes in lakes in different periods (Figure 5). The elevation difference of the lakes in the area is 1240 m, ranging from 4340 m at the lowest to 5580 m at the highest. The study area was divided into six elevation ranges to account for the impact of various Pleistocene glacial periods: <5000 m, 5000 m–5100 m, 5100 m–5200 m, 5200 m–5300 m, 5300 m–5400 m, and <5400 m. Table 3 lists the number, area, area increment, and the number of lakes with or without area increase for each type of lake by elevation range. The distribution of different types of lakes and their elevations are also shown in Table 3. The moraine-dammed lakes

are evenly distributed in all elevation ranges, and the cirque lakes are concentrated below 5000 m. U-type valley lakes cover a range of 5000 m to 5300 m, with most distributed between 5100 and 5200 m. Glacial erosion lakes are distributed in two fields, below 5000 m and between 5100 m and 5300 m. Lateral moraine lakes are concentrated between 5000 m and 5100 m. Figures 6 and 7 show the changes in the number and area of lakes at different elevations. As the elevation increases, it is observed that the area forms a single-peak distribution and reaches a maximum between 5000 m and 5400 m. The number of lakes in the area also increased with the elevation.

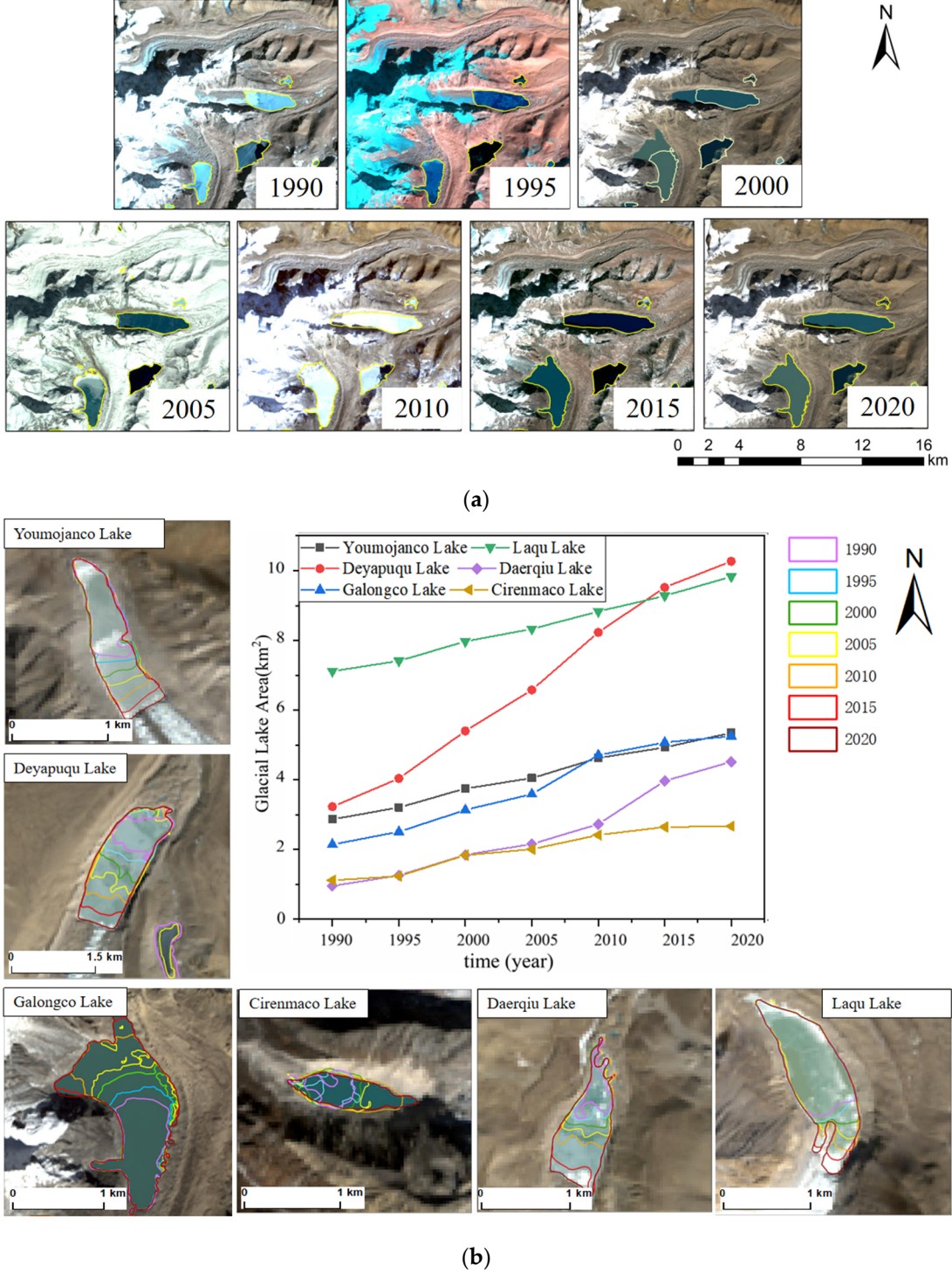

**Figure 5.** (**a**,**b**) Thirty-year comparison of glacial lake area changes in Nyalam County.

**Table 3.** Change of glacial lakes for different elevations.

| Altitude (m) | Current (2020) Total Area (km²) | Moraine-Dammed Lake | U-type Valley Lake | Cirque Lake | Glacial Erosion Lake | Lateral Moraine Lake | Total Area of Increase (km²) | Number of Lakes Showing Areal Increases | Number of Lakes Showing No Increase in Area |
|---|---|---|---|---|---|---|---|---|---|
| <5000 | 1.44 | 6 | 2 | 1 | 1 | | 0.314 | 3 | 7 |
| 5000–5100 | 7.91 | 3 | 1 | | 1 | 1 | 3.017 | 5 | 1 |
| 5100–5200 | 3.45 | 5 | 3 | 2 | | | 0.106 | 7 | 5 |
| 5200–5300 | 7.51 | 12 | | 1 | | | 2.384 | 10 | 3 |
| 5300–5400 | 14.31 | 14 | | 1 | | 1 | 2.263 | 14 | 2 |
| >5400 | 1.109 | 8 | | | | | 0.006 | 2 | 6 |

## 4.1.2. Characteristics of Glacial Lake Changes in Different Periods

Figure 6 shows that the glacial lakes in Nyalam County have expanded significantly in the past three decades. Among them, Figure 6a–c is the highlight of the local area of GongCuo Lake. The surrounding glaciers have gradually receded with time, the number of glacial lakes has increased, and the area has developed. Most of the increased glacial lakes are located at an elevation of 5000 m, all of which are glacial moraine lakes formed by the retreat of glaciers. At the same time, it can be seen from the changes that glacial retreat is the direct cause of the increase in the number and area of glacial lakes [52].

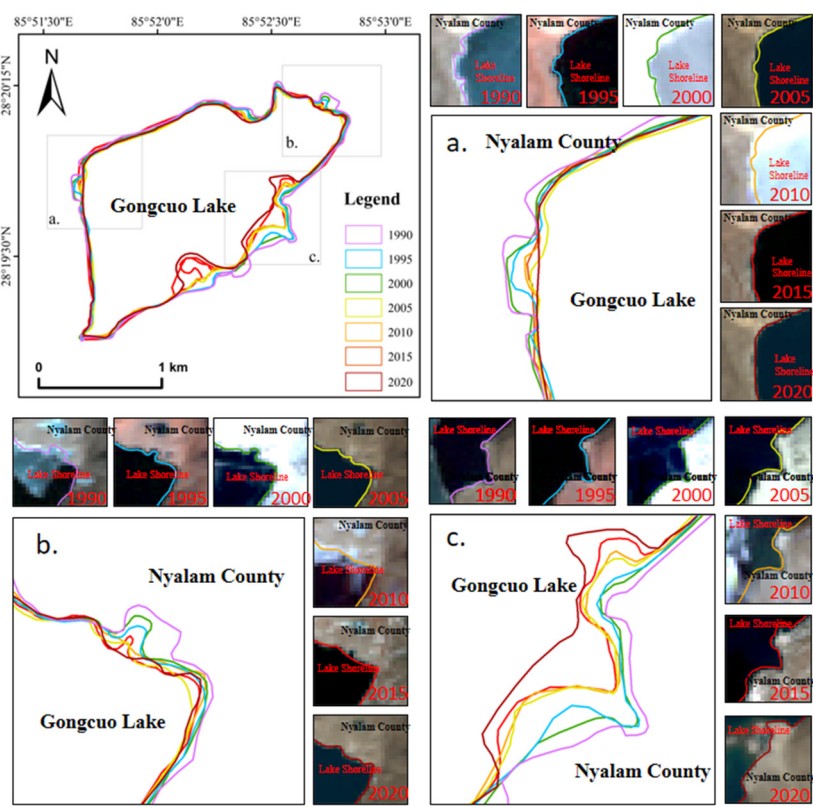

**Figure 6.** Thirty-year area change of glacial lake in Nyalam County ((**a–c**) GongCuo Lake).

Based on historical glacial lake outburst data and visual interpretation [53,54], the distribution characteristics and types of glacial lake outbursts in Nyalam County were extracted and analyzed. Among them, Figure 7a,b is the highlight of the local area of GangxiCo and YinreCo Lake. The loss of glacier mass balance can quantify the information of glacial avalanche hazards [55]. The primary trend in glaciers in the Himalayas and the central and southern Tibetan Plateau is overall mass loss [56]. The southern mountain range's edge was where Nyalam County's ice avalanche disasters were mainly concentrated. Many glaciers in the region create conditions for glacial avalanche disasters [57]. Glacial lakes more prone to outbursts are glacial moraine lakes that end close to supply glaciers [35]. (More details in Supplementary Materials Table S2 [58])

High-resolution remote sensing images in 2018 and 2019 were extracted (Figure 8a,b). Based on the above remote sensing images of GangxiCo glacier lake and its surroundings in Nyalam County and data from the GF-1 satellite, compared with 2018, the area of glacial lakes in 2019 increased by 35,642 m$^2$. Compared with the 107% expansion (0.34 km$^2$/year) of GangxiCo Lake from 1974–2014 [59], the area of the GangxiCo glacier lake expanded faster. When the storage capacity of the glacial lake was calculated according to the relationship between the area and volume, it was found as V = 0.104A$^{1.42}$. The water storage capacity of the glacial lake in 2018 was 301,724,326 m$^3$, and in 2019 the water storage capacity of the glacial lake was 305,041,629 m$^3$, an increase of 3,317,303 m$^3$ compared to 2018.

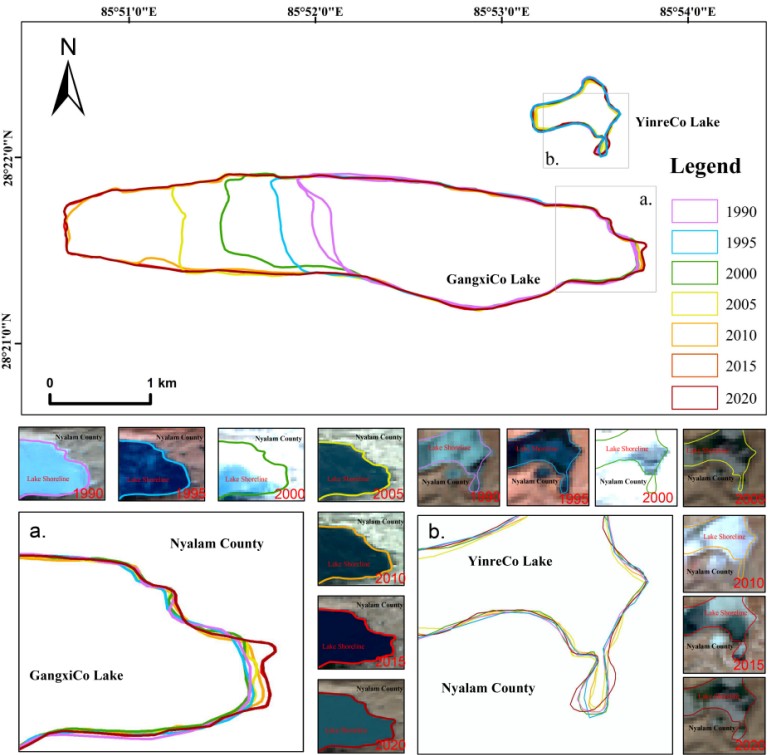

**Figure 7.** Thirty-year area change of glacial lake in Nyalam County ((**a**,**b**) GangxiCo and YinreCo Lake).

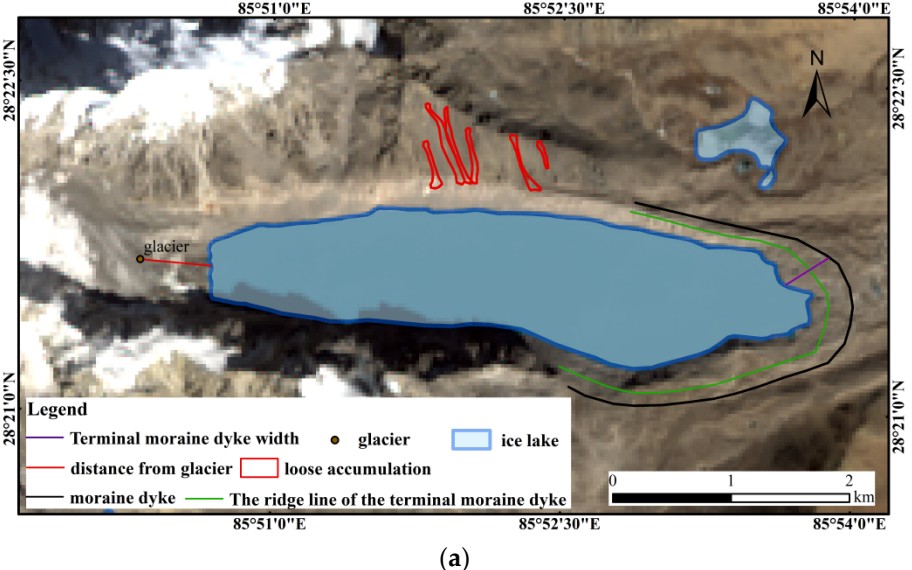

(**a**)

**Figure 8.** *Cont.*

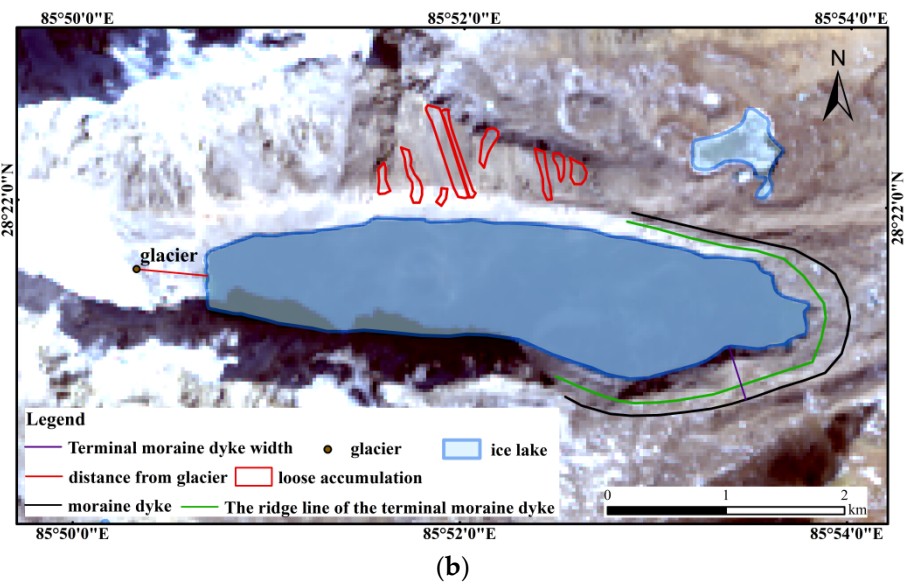

**(b)**

**Figure 8.** (**a**). Interpretation map of GangxiCo glacial lake and surroundings in 2018. (**b**). Interpretation map of GangxiCo glacial lake and surroundings in 2019.

### 4.1.3. Precision Test Results

The calculation result of the Kappa coefficients is −1~1, but the results are usually between 0 and 1. They can be divided into five groups to indicate different levels of consistency: 0.0~0.20 (very low consistency), 0.21~0.4 (average consistency), 0.41~0.6 (moderate consistency), 0.61~0.80 (high consistency), and 0.81~1 (almost completely consistent). Matlab calculations in this paper show (Table 4) that the classification extraction and superposition algorithm are used to extract the glacial lakes in the three states. The misclassification and omission errors of the extracted glacial lake boundary under the three states are less than 2%. The kappa coefficient is more than 0.81, which indicates that the consistency degree of the glacial lake boundary extracted by the classification extraction and superposition algorithm achieves the best results.

**Table 4.** Evaluation of the accuracy of the glacial lake in different states.

| Status | Misclassification Error (%) | Omission Error (%) | Kappa Coefficient |
|---|---|---|---|
| Half icing | 0.67 | 1.40 | 0.86 |
| Total icing | 0.55 | 1.08 | 0.90 |
| Unfrozen | 0.23 | 0.86 | 0.92 |

### 4.2. Analysis Results of Climate Response to Glacial Lake Changes

The expansion of the glacial lake in Nyalam is closely related to external climate change. This paper analyzes the glacial lakes' evolution characteristics statistically sorted by periods. The average statistics of climate factors are calculated in each decade as a period, and the climate change rate in different periods is calculated by linear fitting based on the annual data. By sorting out and calculating the changes in the glacial lake area and the changes in the corresponding climatic factors, it is found that the GangxiCo in Nyalam County has expanded by a total of 1.88 km. Among them, the area of glacial lakes developed by 0.699 km$^2$, 1.051 km$^2$, and 0.13 km$^2$, respectively, in 1990–2000, 2000–2010, and 2010–2020. The statistics of the growth area and the average value of meteorological factors in the three periods are shown in Table 5. The nearest observed meteorological station in the Nyalam area to the glacial lake, Nyalam station, was selected (85°58′E, 28°11′N).

**Table 5.** Thirty-year annual change in Nyalam County.

| | 1990–2000 | 2000–2010 | 2010–2020 | 1990–2020 |
|---|---|---|---|---|
| Glacial lake growth area (km²) | 0.699 | 1.051 | 0.13 | 1.88 |
| Annual temperature (°C) | 3.69 | 4.312 | 4.158 | 4.042 |
| Summer temperature (°C) | 10.2919 | 10.6067 | 10.4191 | 10.441 |
| Winter temperature (°C) | −2.897 | −1.673 | −1.987 | −2.2 |
| Annual precipitation (mm) | 551.76 | 613.08 | 632.93 | 595.51 |
| Summer precipitation (mm) | 203.63 | 198.86 | 220.95 | 206.36 |
| Precipitation in winter (mm) | 99.72 | 102.79 | 141.1 | 111.6 |
| Sunshine hours (h) | 2515.49 | 2477.95 | 2507.76 | 2499.58 |
| Average wind speed (m/s) | 4.686 | 4.063 | 4.069 | 4.29 |
| Average relative humidity (%) | 67.39 | 67.71 | 62.9 | 66.34 |

### 4.2.1. GeogDetector Model Analysis Results

GeogDetector conducted an environmental climate impact factor analysis on the occurrence (Y) of glacial lake areas in Nyalam County from 1990 to 2020. The climate impact factors or proxy variables (X) include sunshine hours, average temperature, wind speed, relative humidity, and precipitation. Table 6(a) shows the risk factor interaction detection results, where rows 2~5 show the q-values after the interaction of two variables. The results show that the exchange of any two variables on the spatial distribution of the glacial lake is greater than that of the first variable alone, reflecting the results of the risk area detection for a single risk (climate impact factor).

**Table 6.** (a). Results of cross-detection. (b). Ecological detection and risk factor detection.

**(a) Interaction_detector**

| | Sunshine hours | Average temperature | Wind speed | Relative humidity | Precipitation |
|---|---|---|---|---|---|
| Sunshine hours | 0.082 | | | | |
| Average temperature | 0.270 | 0.073 | | | |
| Wind speed | 0.492 | 0.248 | 0.193 | | |
| Relative humidity | 0.559 | 0.557 | 0.606 | 0.451 | |
| Precipitation | 0.531 | 0.558 | 0.732 | 0.677 | 0.342 |

**(b) Sig. F test: 0.05**

| | Temperature | Precipitation | Humidity |
|---|---|---|---|
| Temperature | | | |
| Precipitation | Y | | |
| Humidity | Y | Y | |
| q statistic | 0.997703117 | 0.999494156 | 0.994113327 |

In the first row of the table in Table 6(b), "temperature", "precipitation", and "humidity" are the names of each partition of this environmental factor, which are the type quantities. The numerical quantities in the second row are the change in the area within each type zone. The following rows 3~4 are for determining whether there is a statistical difference between the changes of a glacial lake in each type of partition, using the t-test with a significance level of 0.05, and "Y" indicating significant differences between "temperature", "precipitation" and "humidity".

Table 6(b) shows the calculation results of q-values for all risk factors. The results indicate that precipitation variables, temperature, and humidity have high q-values, meaning that rivers among these variables are the main environmental factors that determine the glacial lakes' spatial pattern.

### 4.2.2. Multivariate Linear Regression Analysis Results

Changes in glacial lakes are closely related to global climate change. Glaciers recede when the weather warms up, supplying high lakes with meltwater as the lakes grow in

size [60,61]. Seepage lubricates the glacier's bottom during severe melting, which may lead to frontal collapse if the glacier extends into the lake, thus triggering lake eruption. The expansion of TP glacial lakes can be explained by the trend of increased precipitation and accelerated melting of glaciers associated with rising temperatures [62]. Many studies have provided substantial evidence for lake and glacier dynamics on the TP under climate fluctuations [63]. Some researchers suggest that increased regional precipitation between 1990 and 2010 led to a significant expansion of enclosed lakes on the TP [64,65]. At the same time, others argue that the expansion of lakes fed by glaciers is primarily caused by rising temperatures that accelerate glacier shrinkage [52,66]. Based on the above research experience, this paper selects these climate-influencing factors for correlation analysis. This paper uses data from 1990 to 2020 from meteorological stations in Nyalam County based on five climate impact factors: annual average precipitation, annual average temperature, annual average sunshine hours, annual average relative humidity, and annual average wind speed. Figure 9 shows the statistical data, which contain the annual average temperature of the Nyalam region and 10-year and 30-year average data, and a linear trend line composed of extremely monthly averages. The figure shows that the temperature does not vary much yearly, and the trend is slowly increasing. The temperature increased by 0.91 °C, as demonstrated by the 30-year average temperature of 4.04 °C, and the temperature rapidly increased from 2005 to 2010, and the average temperature was significantly higher than the overall average temperature. The average temperature shows a general warming trend, with a temperature increase of 1.5 °C and a temperature increase rate of 0.6 °C (10 years), which is higher than the national temperature increase rate ((0.26 ± 0.032 °C)/10 years). Precipitation in different years is significantly different, with an overall upward trend and an increase of 81.17 mm, and a precipitation growth rate of 27.06 mm (10 years). The average annual precipitation varies: 30-year average precipitation is 595.51 mm. The average precipitation from 1990 to 2000 is 551.76 mm, which is 43.75 mm less than the overall observations. After 2000, the value began to fluctuate around the linear trend line.

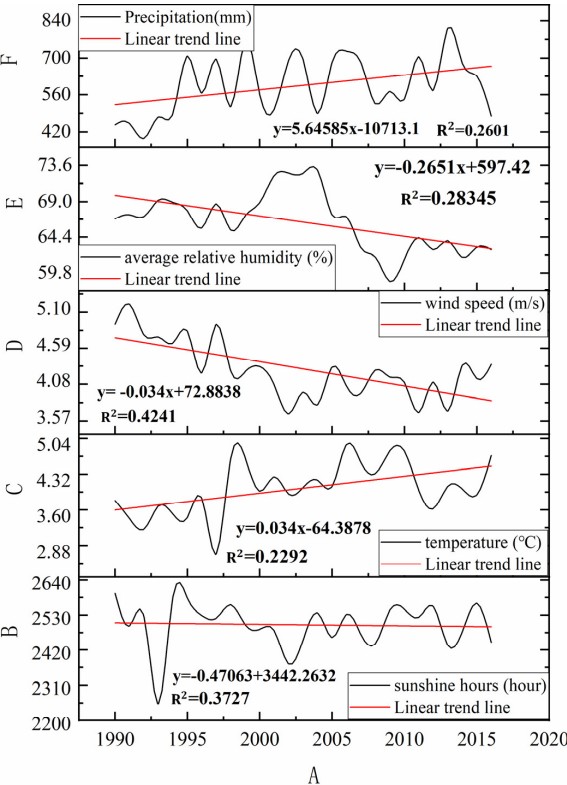

**Figure 9.** The thirty-year trend of climate factors. (**A**), year, (**B**), sunshine hours, (**C**), temperature, (**D**), wind speed, (**E**), average relative humidity, (**F**), precipitation.

A tolerance of less than 0.1 indicates severe multicollinearity in the covariance statistics, and a variance expansion factor VIF, which was more significant than 10, indicates severe collinearity. When the VIF value is less than 10, we believe that the data conform to the multiple linear analysis, and there is no multicollinearity among the independent variables. It can be seen from the table that the VIFs of sunshine hours, average temperature, average wind speed, average relative humidity, and precipitation are all less than 10, and their tolerances are all close to 0.8. It can be stated that there is no multicollinearity problem in this model. Analysis of variance (ANOVA) is a significant test invented by R.A. Fisher for the difference in the mean of more than one sample. This paper uses ANOVA to judge whether the regression equation is essential. The corresponding *p*-value in the ANOVA model, which is the sig in the table, is significant and indicates the size of the difference between the control and the experimental group. In the table, $p = 0.011 < 0.05$ suggests that the null hypothesis is supported. That is, the linear regression equation is significant. The corresponding T-test in Table 7 is to judge the importance of each variable of the regression equation.

**Table 7.** Table of coefficients.

| | Model | Unstandardized Coefficient | | Standardized Coefficient Trial Version | t | Sig. | Covariance Statistics | |
|---|---|---|---|---|---|---|---|---|
| | | B | Standard Error | | | | Tolerance | VIF |
| 1 | (Constant) | 163.126 | 46.453 | | 3.512 | 0.002 | | |
| | Sunshine hours | −0.010 | 0.012 | −0.134 | −0.825 | 0.419 | 0.926 | 1.080 |
| | Average temperature | −2.462 | 2.180 | −0.239 | −1.130 | 0.271 | 0.546 | 1.833 |
| | Average wind speed | −6.922 | 3.013 | −0.513 | −2.298 | 0.032 | 0.490 | 2.040 |
| | Average relative humidity | −0.961 | 0.256 | −0.660 | −3.758 | 0.001 | 0.791 | 1.264 |
| | Precipitation | −0.006 | 0.009 | −0.122 | −0.674 | 0.508 | 0.746 | 1.341 |

Coefficients [a]

[a] Dependent variables: area.

From the above calculation, the optimized regression equation is:

$$Y = -0.010x_1 - 2.462x_2 - 6.992x_3 - 0.961x_4 - 0.006x_5 + 163.1260 \tag{21}$$

The goodness of fit $R^2$ is 0.689, and the regression coefficients of all independent variables passed the t-test, with an excellent overall fit and no multicollinearity. The criterion of standardized residuals by regression illustrates that this study's polynomial linear simulation model meets the regression requirements better.

Figure 10 shows that the significance ranking of the variation of a glacial lake in the Nyalam County area is mean relative humidity > mean temperature > sunshine hours > precipitation > mean wind speed. Overall, the average annual climate is mainly influenced by the winter temperature. The yearly average temperature from 1990 to 2020 shows a slowly increasing trend with a climate tendency rate of 0.34 °C (10 years). The annual precipitation shows a weak (insignificant) increasing trend, and the climate trend rate is 5.64 mm/(10 years). The yearly temperature is positively correlated with the change in the glacial lake area, and the annual precipitation is negatively correlated with the difference in the glacial lake area (r = −0.23). However, the glacial lake area change is highly associated with summer precipitation (r = −0.57), indicating that the area varies with the seasons. Precipitation is also one of the important reasons for the change of glacial lakes. Since the humidity is highly correlated with the area change (r = 0.56), the variance of the average relative humidity and the sunshine change obtained by the univariate linear model is slight, and the annual difference is not apparent. Therefore, the changes in Nyalam County's glacial lakes are mainly affected by temperature and precipitation changes. From 1990 to 2020, the temperature and precipitation in Nyalam County show an increasing trend. Temperature and precipitation are meteorological factors with high correlation. The

continuous increase in temperature is the dominant factor in the melting of glaciers, and the melting of glaciers has caused an increase in the area of glacial lakes. Continued increases in temperature cause glaciers to retreat and produce meltwater, increasing the size of glacial lakes. At the same time, it is also affected by the topography and the temperature rise in summer and winter to varying degrees, and the humidity and sunshine hours also change accordingly. Global warming is essential in the growth of glacial lakes in Nyalam County (more details in Supplementary Materials Figure S2).

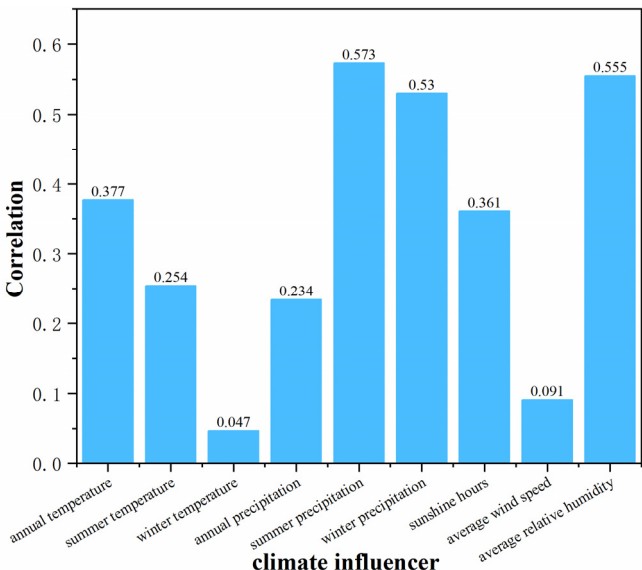

**Figure 10.** Correlation of climate factors.

### 4.2.3. Mann–Kendall Trend Test and Estimation of Sen's Slopes Results

As mentioned before, in addition to the correlation analysis, we also estimated the slope of Sen to capture the magnitude of change in each lake with climatic variables. This paper uses the Mann–Kendall test method for significant trend analysis. At the same time, the glacial lake area data from 1990 to 2020 were imported, and the standardized test statistic Z was defined by calculation. When the absolute values of Z were more significant than 1.65, 1.96, and 2.58, the trend passed the significance test with 90%, 95%, and 99% reliability, respectively. After testing, Z = 2.73 > 2.58, which indicates that there is an apparent trend change in the data. In addition, $p = 0.007 < 0.05$, meaning that the time series data have a trend. In addition, the difference in the Sen slope indicates that the glacial lake area is gradually increasing (Sen's slope = 4.16). Finally, the Mann–Kendall test examined the importance of trends. All trends related to the glacial lake area were significant, and the Mann–Kendall test shows important trends ($p \leq 0.05$) for wind speed, air temperature, and humidity. In addition, the change in Sen's slope shows that depending on the lake, although the temperature has increased significantly (Sen's slope = 0.34), the volume of some lakes may decline or remain the same. This difference may imply that the rates of lake area change are not the same under the influence of climatic factors, so it is worth investigating the relationship between changes in lake variables and geography, and other lake characteristics. Furthermore, there are many factors to consider, such as lake morphology, changes in frozen glacier soil, and surrounding glaciers.

To investigate the impact of temperature and glacier-frozen soil on the area change of glacial lakes, this paper sealed the soil samples of the glacial lake and moisture content, wet density, dry density, porosity ratio, saturation, soil particle specific gravity, and particle size analysis on the soil samples. First, the essential engineering–geological parameters of the samples, such as the maximum, minimum, average value, standard deviation, coefficient of variation, etc., were calculated according to the routine analysis of the laboratory test results. Then, according to the test analysis results, the particle size curves of each soil

sample were drawn, as shown in Figure 11a–d. After arranging the data in units of test pits, the frequency map of each test pit sample's particle size distribution, as shown in Figure 11e,f, was obtained. It can be seen that the particle size of glacial lake sediment particles is distributed in multipeaks between 0.005~0.075 mm and 2~10 mm, with poor sorting and extremely poor rounding, which are typical glacial sands.

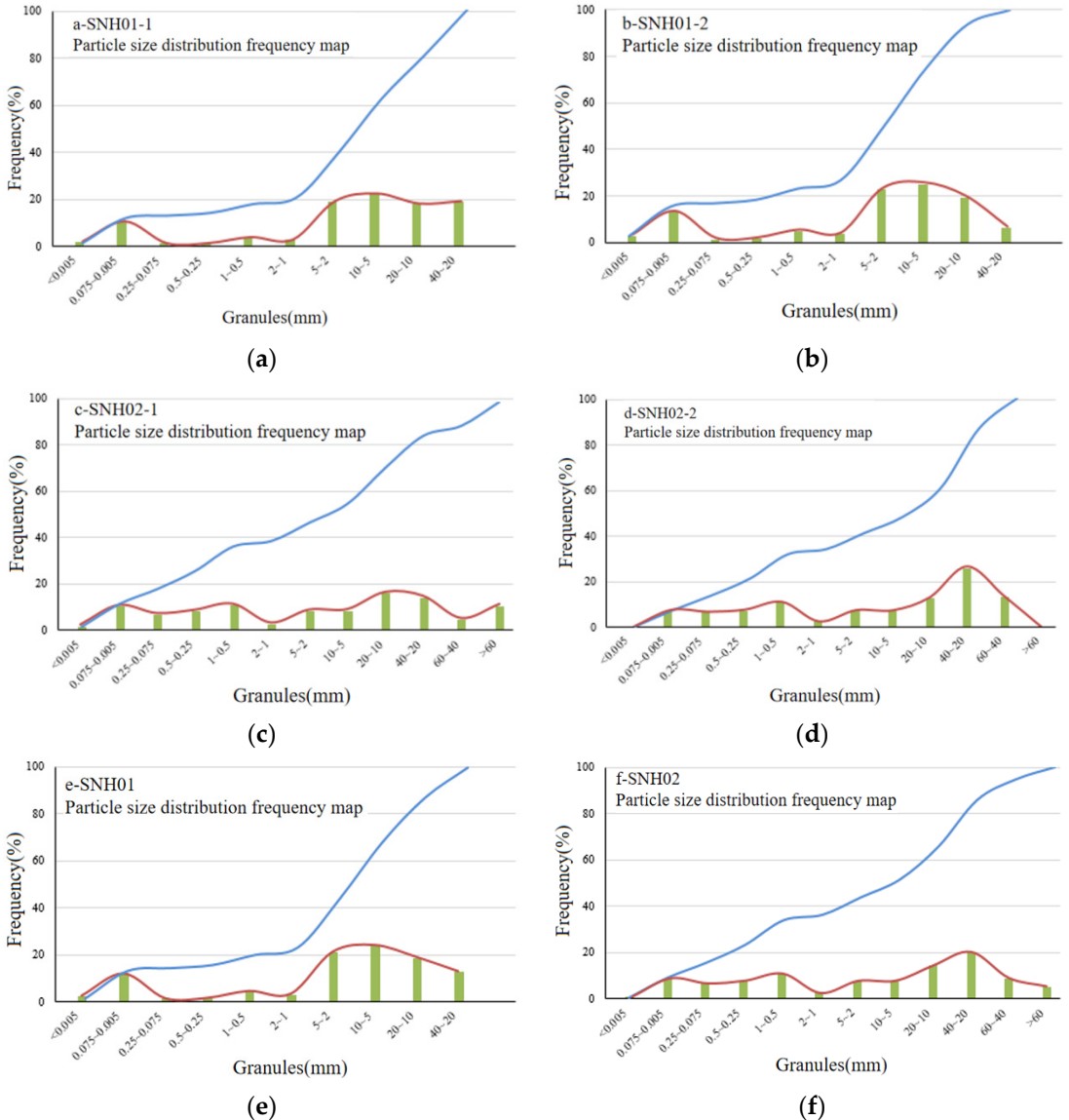

**Figure 11.** Particle size profile of glacial lake soil sample (**a**), SNH01-1, (**b**), SNH01-2, (**c**), SNH02-1, (**d**), SNH02-2, (**e**), SNH01, (**f**), SNH02.

Since the water level of the glacial lake between 2019 and 2020 was highly similar to the temperature trend, it directly shows an increase in precipitation and a decrease in lake evaporation, indicating that glacial melting was likely the leading cause of glacial lake expansion in the region. Because of the strong correlation between temperature and water volume, most glacial lake sediment particles belong to typical glacial sands. Therefore, this paper infers that the increase in temperature not only melts the glacier but also causes the glacier's frozen soil to thaw.

### 4.3. Simulation Results of a Glacial Lake Outburst Event

Based on DEM in the simulation, the physical input parameters of HEC-GeoRAS were obtained through ArcGIS analysis. The range of calibrated Manning resistivity n in the

main channel was obtained using the nominal curve values exercise. The simulation results were exported to ArcGIS to depict the water surface in the floodplain. The maximum flow values from the observatory were used for the upstream flood input to simulate a glacial lake outbursts scenario. The measured water surface elevation was 5181.01 m, the contact length between the water crossing section and the water body was L = 17.19 m, and the water depth was h = 0.15 m. Taking these parameters into the Manning formula, the flow rate of GangxiCo is about 0.67 m$^3$/s during this measurement period. Compares the images of the glacial lake before and after the breach (Figure 12). At this time, the maximum storage volume can be calculated as 431,042 × 10$^3$ m$^3$. The flood inundation area of HEC-RAS in the simulated river reach (402.53 m) is about 42.87 m$^2$. In the top-left diagram, the line inside the lake is the isobath, and the line outside the lake is the Contour line.

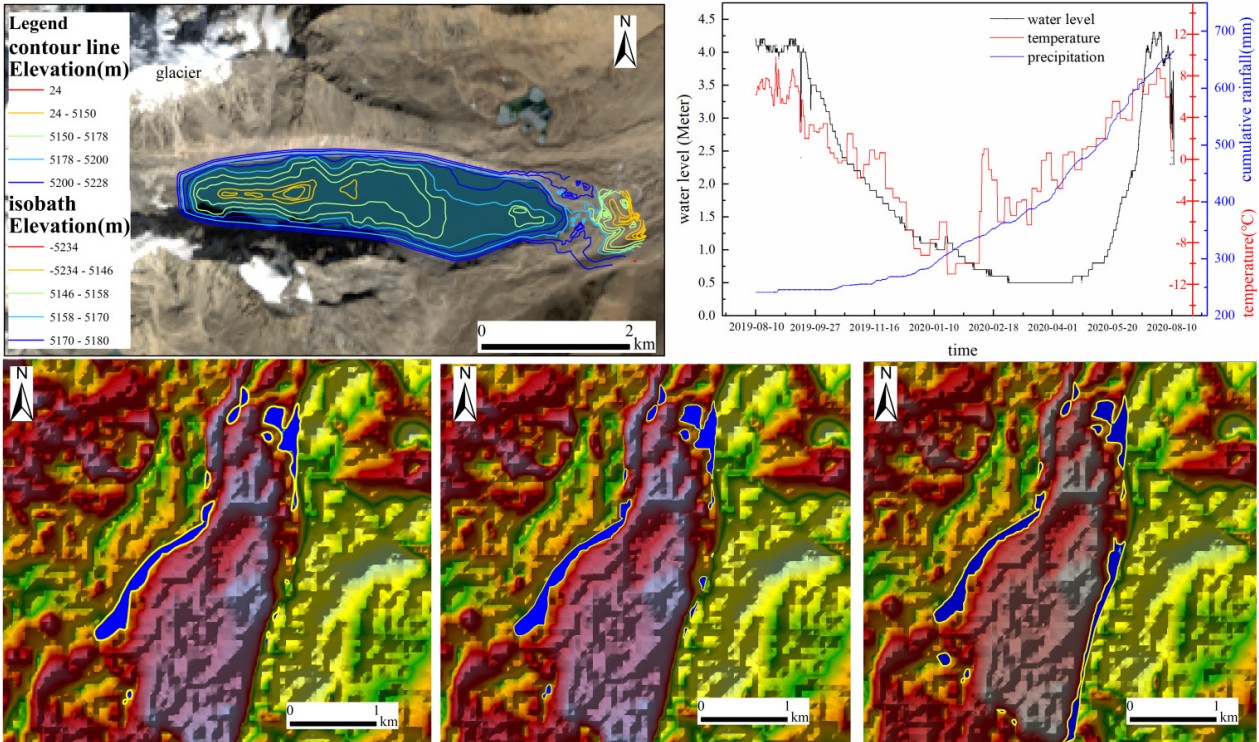

**Figure 12.** Changes before and after the glacial lake outburst.

The overall pattern of glacial lake extension over the last three decades is obvious. The GangxiCo lake's current change rate has been quite gradual, and the estimated value of the lake evolution rate under climate change is 0.52% (0.02 km$^2$·a$^{-1}$). GangxiCo Lake is now in equilibrium and showing a slightly growing tendency [67,68]. Combined with the stable changes in the dam structure and the nearby glaciers, there is no risk of dyke breach in the GangxiCo glacial lake. However, the glacial lake in Nyalam County is susceptible to changes in climate factors. It is not only affected by highly correlated climatic factors such as precipitation, humidity, and temperature, but the surface water temperature and wind speed are important factors affecting the evaporation of the glacial lake. These factors have different degrees of influence in various areas of Nyalam County. The monitoring and simulation of surface water temperature in Namtso and GangxiCo show that the surface water temperature of the glacial lake increase with the increase in temperature and long-wave radiation. We can infer the corresponding climate factors that cause the changes in glacial lakes through the changes in glacial lake water volume. Hence, future estimates are further analyzed according to the predicted precipitation: from 2016 to 2025, the rate of climate change may continue to the current rate, and the lake area will continue to increase, while from 2026 to 2035, the rate of evolution of the lake may increase as the

climate is expected to become moister and the risk of glacial lake outbursts can gradually increase [69].

## 5. Discussion

The particular geographical environment of the Qinghai–Tibet Plateau leads to the freezing of glacial lakes in most cases. In contrast, most existing methods aim to extract nonfrozen lake boundaries. The frozen lake's spectral features are similar to those of glaciers [70]. Therefore, this paper uses the classification extraction superposition method to extract the glacial lake boundaries. According to the geographical conditions and glacial lake distribution in Nyalam County, the classification method of classification extraction superposition of glacial lake information is proposed under a stepwise iterative extraction algorithm for remote sensing information. This method overcomes the interference of common unfavorable factors such as glaciers and mountain shadows on glacial lake information extraction. It effectively improves glacial lake information extraction detection speed and quality. The glacial lake is classified as frozen and unfrozen, and the NDWI method is used to extract the unfrozen glacial lake boundary. The frozen glacial lake boundary is extracted using the difference between the frozen surface in the red band and NIR, which are more significant than the appropriate threshold and the red bar is more effective than a specific value. Finally, the unfrozen and frozen glacial lake boundaries are superimposed to obtain the complete glacial lake boundary. Some studies used object-oriented image-processing methods to extract glacial lakes. This paper proposes an innovative analysis method for classification and stacking extraction, which improves the extraction accuracy (Kappa coefficient reaches 90%). This method can perform spatiotemporal analysis on a cloud platform with high speed and efficiency. It can overcome the shortcomings of incomplete information, long processing time, and repetitive tasks and improve the extraction efficiency of long-sequence glacial lakes. At the same time, the classification stack extraction method proposed in this study has the potential to be extended to other debris-covered glacial environment observations. However, since this paper's extraction method for unfrozen glacial lakes is based on the traditional NDWI method, the extraction of unfrozen glacial lakes is limited mainly by this method, especially in areas with severe snowfall in winter. When snow accumulates on the glacial lake, it is difficult to distinguish the specific boundary of the glacial lake, and the threshold needs to be adjusted according to the regional characteristics. The method is subsequently improved to obtain a better way of extracting the boundaries of glacial lakes.

Previous studies reported that glacial lakes began to expand with climate warming but did not specifically measure the changes and attribute the associated drivers [71]. In this paper, the characteristics of climate change and glacial lakes' temporal and spatial changes are coupled for analysis. When analyzing the driving factors of glacial lake area changes, it is considered that there may be correlations between influencing factors and the interaction of influencing factors on dependent variables. The geographic detector measured the climatic factors affecting the area change of the glacial lake, and then the multiple linear regression analysis was carried out. The response mechanism of glacial lakes in the Nyalam area to climate change and the differences in response mechanisms between different regions are revealed.

Additionally, a combined analysis of the changes in glacial lake area, water volume, and climate conditions was conducted to inform the future changes. Based on the monitoring of the water level, the flow of the glacial lake in Nyalam County, and the analysis of the corresponding climate, it was found that the lake showed a significant expansion between 1999 and 2010. Because the change in the glacial lake area depends on the balance of water revenue and expenditure [72], it is closely related to climate change [73]. The warm-season temperature increase will replenish glacier and seasonal snow meltwater to the glacial lake and promote the evaporation dissipation of the lake water when the evaporation of the glacial lake is less than the recharge of the glacial lake, making the glacial lake area decrease. The increase in cold-season temperature may cause the ice and snow temperature

to increase to some extent, accelerating the melting rate of ice and snow during the warm season. The increase in precipitation also means the recharge of the glacial lake directly or indirectly by snowmelt through slope runoffs. Among them, the water level of the glacial lake from 2019 to 2020 is highly similar to the temperature trend, which means that the temperature change is likely to cause the melting of the glaciers. This melting is the main reason for the region's expansion of the glacial lake because of the strong correlation between temperature and water volume and the multipeak distribution of particle size in glacial lake sediment. This paper infers that the temperature increase is not only the leading cause of glacial melting but also the thawing of frozen soil. This provides a preliminary research conclusion with specific reference significance for studying glacial lakes in the area. In terms of glacial lakes' spatiotemporal characteristics and their response to climate, this study mainly used mathematical analysis methods such as linear fitting, numerical comparison, and multivariate joint analysis for analysis and comparison. Methods are complex, and it is also necessary to introduce confidential data to reveal temporal and spatial evolution of the glacial lakes from a deeper dimension.

Recent studies on the hazard/risk assessment of glacial lakes in the Himalayas and some distinct regions have demonstrated a high degree of consistency and consensus in evaluating high-risk glacial lakes [74], even though assessment protocols vary [75,76]. Current research on glacial lakes mainly focuses on the distribution characteristics of contemporary or potential glacial lakes and their accompanying hazards or risks [77]. However, research on prevention and mitigation measures for GLOF from high-hazard/-risk glacial lakes is still lacking [78,79]. This paper provides a new model for evaluating GLOF risk from water volume variability and associated climate impact. In addition, it fills the gap in information about glacial lake outburst data in the Nyalam region using HEC-RAS software. The two-dimensional unsteady flow numerical model proposed in this paper provides a quantitative method for studying the characteristics of glacial lake outburst floods. The primary control equation is based on a complete two-dimensional water balance. The HEC-RAS software applied a numerical solution that captures the topography and automatically handles complex boundaries.

Furthermore, the numerical model has good performance in handling complex topography for simulating glacial lake outburst flood processes [80]. The HEC-RAS uses a parametric model to predict the break parameters. Simultaneously, it introduces a simplified physical dam break mechanism for the development of the break to the final break while solving the process of the break outflow. The model can be applied to the outburst simulation of small and medium glacial lakes by introducing a physical outburst model. This paper analyzes the impact of climate conditions on the GLOF outburst mechanism, combined with glacial lake changes and related climate changes over the past three decades, including the characteristics of the outburst formation process, duration, and flood peak flow. The threshold is set to provide experimental data for the future prevention and control of the area, which is of great significance for the areas at risk of glacial lake outbursts with the increasing medium–low flow. However, this paper mainly monitors and examines factors such as quantity, area, and water volume to analyze glacial lakes' spatial structure. Considering the lack of analysis of the overall system formed by the glacial lake and nearby glaciers, more methods need to be introduced to analyze the structural stability of the whole glacial lake system.

## 6. Conclusions

The study of glacial lakes is significant due to their sensitivity and particularity to inform climate change. However, remote sensing of glacial lakes in southern Tibet faces the challenges of covered glacial environments and hillsides. In this study, we proposed an improved estimation method for the area distribution and development of glacial lakes and an innovative approach for extracting glacial lakes by classification and stacking, reducing the impact of hillsides and surrounding glaciers on extraction precision. We also analyzed glacial lake changes and outbursts using long-term imageries that combine meteorological

and bathymetric data. Our studies illustrated that: (1) The area of glacial lakes increased sharply from 1990 to 2020, but the number of glacial lakes did not increase simultaneously. The increase in the area of glacial lakes has a unimodal distribution, mainly concentrated in the range of 5100–5400 m above sea level. (2) The continuous increase in temperature leads to the long-term retreat of glaciers and the production of meltwater, which increases the area of glacial lakes and has a far-reaching impact on the growth of glacial lakes in Nyalam County. The study found that glacial lake sediments are typical of glacial sands, and rising temperatures help melt glaciers, which is the primary reason for the thawing of frozen soil. (3) Through mathematical simulation, the maximum water volume that the GangxiCo can withstand is $431,042 \times 10^3$ m$^3$. The introduction of the classification extraction stacking algorithm and the outburst analysis of the HEC-RAS model provides a new perspective for remonitoring the environmental dynamics of glacial lakes in climate change. This method may be extended to the outburst simulation of other glacial lakes. The future trend can be projected by comparing the increase and maximum water volume of glacial lakes with glacial meltwater and precipitation in the future, providing accurate and reasonable data for regional flood control and disaster reduction.

**Supplementary Materials:** The following supporting information can be downloaded at: https://www.mdpi.com/article/10.3390/rs14194688/s1, Figure S1: Glacial lake area change; Figure S2: Standard P-P plot of regression standardized residuals; Table S1: Glacial lake change Information; Table S2: Since the 20th century, the outburst of the Glacier Lake in Nyalam County, Tibet, China; Table S3: The Confusion Matrix for the extraction accuracy of Glacial Lake.

**Author Contributions:** Conceptualization, X.D.; data curation, G.Q., J.C., W.L. (Weile Li) and Z.Y.; funding acquisition, X.D. and W.L. (Weile Li); methodology, G.Q., X.D. and J.C.; resources, G.Q., X.D., W.L. (Weile Li), Z.Y. and Y.S.; software, G.Q. and W.L. (Wenxin Liu); validation, G.Q., W.L. (Wenxin Liu) and Y.S.; visualization, G.Q., X.D., Y.S. and J.R.; writing—original draft, G.Q., X.D. and B.Z.; writing—review and editing, G.Q., J.C., M.W., W.L. (Wenxin Liu), Z.Y., H.L., Y.W. and M.A. All authors have read and agreed to the published version of the manuscript.

**Funding:** This research was funded by the National Key Research and Development Program of China, grant No. 2021YFC3000401; the Research Center for Human Geography of Tibetan Plateau and Its Eastern Slope (Chengdu University of Technology), grant No. RWDL2021-ZD003; Key Research Bases of Humanities and Social Sciences in Higher Education in Sichuan Province, Sichuan Center for Disaster Economic Research, grant No. ZHJJ2021-ZD001 and the National Key Research and Development Program of China under Grant 2017YFB0503601.

**Data Availability Statement:** Not applicable.

**Acknowledgments:** We acknowledge any support given which is not covered by the author's contributions or funding sections. Includes administrative and technical support that is not covered, as well as data materials for experiments.

**Conflicts of Interest:** The authors declare no conflict of interest.

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
