# Peer review of "Characterization of Long-Time Series Variation of Glacial Lakes in Southwestern Tibet: A Case Study in the Nyalam County"

_remotesensing, doi:10.3390/rs14194688_

Round 1

Reviewer 1 Report (Previous Reviewer 1)

The authors made significant improvements to the manuscript quality. They added missing items and corrected descriptions. In my opinion, the article can be published.

Author Response

Reviewer 2 Report (New Reviewer)

The manuscript is devoted to temporal variations of glacial lakes in Southwestern Tibet on relatively short time scale – 30 years from 1990 to 2020. The authors tried to find correlations between bathymetry of these lakes and meteorological data, such as temperature, precipitation, number of sunshine hours, wind speed, humidity). The study demonstrates rather sufficient increase of lakes area, especially at elevation of 5100-5400 m above sea level. Interesting is that increase of lakes area (almost doubling for 30 years) connected with glaciers melting as well as with growth of precipitation is not followed by increase of lakes number. It demonstrates the relative stability of situation based on land surface topography of the region under study. The results can serve as good background for the future monitoring of the situation in this region.

There are some minor mistakes/misprints in the text – e.g. in the table 6 the line “glacial lake growth area” is doubled. According to expert opinion this manuscript can be accepted for publication after the minor revision.

Author Response

Reviewer 3 Report (New Reviewer)

The research proposes an approach for extracting glacial lakes by classification and stacking, which can reduce the impact of hill shade and surrounding glaciers. After estimating the area of glacial lakes, the research also analyzed the correlation between glacial lake changes and climate change. The research results can be helpful for future research on glacial lakes and monitoring of glacial lake outburst disasters. However, this paper has some minor mistakes which need to be modified.

1. In Introduction, although the author shows the method used in the study combination with the research progress, there are too many parts of the research background and progress, especially in the part of the research progress in extracting information about glacial lakes.

2. One of keywords is synergistic evolution, the content contained in this keyword is not mentioned in Abstract, which may cause some confusion to readers.

3. Part 4.3 can further discuss the approximate value of lake evolution rate under the current climate change.

4. The study is analyzing and researching the outburst of the glacial lake, but it has not made a precise definition of the outburst of glacial lake. Please add the definition to make the paper more complete.

5. The English format of your manuscript need to be improved before resubmission. Such as the format error on line 72, and there are a lot similar error in the manuscript, please carefully correct them.

6. There are some other minor mistakes. Some abbreviations are spelled out in full at the beginning such as “GLOF” in line 114, which may lead to some misunderstandings for readers, please indicating the full name next to the abbreviation at the beginning; Besides, the reference styles are suggested to be double-checked.

I would be very glad to re-review the paper in greater depth once it has been edited because the subject is interesting.

Author Response

This manuscript is a resubmission of an earlier submission. The following is a list of the peer review reports and author responses from that submission.

Round 1

Reviewer 1 Report

The article concerns a classification and stacking extraction method for the extractionof glacial lake boundaries under different states. The subject matter is very interesting and interdisciplinary.

The abstract is written correctly, although it can be added a sentence that will emphasize the novelty of the work.

In the introduction, it is possible to point to other articles that concern similar research, not necessarily related to the analyzed region. The description of the analyzed area should be extended. There is no description of the physical and geographical conditions.

The research methodology is described correctly and sufficiently. The description of the results and the discussion do not raise any objections.

In summary, please add a short description of the applicability of the indicated method to another area of research (limitations, accuracy).

Author Response

Comments and Suggestions for Authors

Comment 1: The article concerns a classification and stacking extraction method for the extraction of glacial lake boundaries under different states. The subject matter is very interesting and interdisciplinary. The abstract is written correctly, although it can be added a sentence that will emphasize the novelty of the work.

Response:Thank you for your suggestions. For the mountainous areas of southwestern Tibet, the method of classification, extraction, and superposition is used to extract the glacial lake accurately. And the use of geographic detectors, based on their advantages of detecting spatial heterogeneity, provides better technical support for further understanding the dynamic changes of glacial lakes. We have rewritten the sentence as follows:(line:27-31)

Abstract: Glacial lakes are necessary freshwater resources and essential factors for natural disasters in southern Tibet. The outburst of floods caused by glacial lakes has significantly impacted local people. Since the changes in glacial lakes are closely related to climate change, it is necessary to conduct long-term sequence change detection research on glacial lakes. We propose an innovative method of classification and stacking extraction to accurately extract glacial lakes in southwestern Tibet in the past three decades from 1990 to 2020. Based on Landsat images and meteorological data, geographic detectors were used to detect correlation factors, Multiple regression models were used to analyze the driving factors of ice lake area changes. We combined bathymetric data from glacial lakes with changes in climatic parameters to utilize HEC-RAS to determine critical circumstances for glacial lake outbursts. The results showed that the area of glacial lakes in Nyalam County increased from 27.95 km2 in 1990 to 52.85 km2 in 2020 and eight more glacial lakes observed in the study area. The glacial lake area expanded by 89.09%, with significant growth from 2015 to 2020. Correlation analysis between the glacial lake area and climate change throughout the period shows that temperature and precipitation dominated the expansion of these lakes from 1990 to 2020. We also discovered that the progressive increase in water volume of glacial lakes is most likely linked to the constant rise in temperature and freeze-thaw of surrounding glaciers. Finally, the critical conditions for the glacial lake's outburst were judged using HEC-RAS combined with the changes in the water volume and climatic factors. It is concluded that GangxiCo endures a maximum water flow of 4.3×108m3, and the glacial lake is in a stable changing stage. This conclusion is consistent with the realistic investigation and can be used to provide scientific guidance for predicting glacial lake outbursts in Southwest Tibet in the future.

Comment 2: In the introduction, it is possible to point to other articles that concern similar research, not necessarily related to the analyzed region.

Response:Thanks for your comments. In response to your comments, we add other articles involving similar studies in the introduction.

Comment 3: The description of the analyzed area should be extended. There is no description of the physical and geographical conditions.

Response:Thank you for your suggestions. We have rewritten the sentence as follows:(line:141-147)

The geomorphology of the study area presents an alternating pattern of alpine valleys and plateau lake basins. Among them, large glacial lakes are formed in a concentrated distribution near the alpine glaciers, and small glacial trough lakes and ice lakes are scatterly distributed in the valleys. The region mainly has a sub-arctic semi-arid climate and a humid subtropical climate. The dry season has a distinct rainy season. The rainy season is in June and August while the snowfall lasts for six months. The huge rainfall is likely to cause the glacial lake outburst.

Comment 4: The research methodology is described correctly and sufficiently. The description of the results and the discussion do not raise any objections. In summary, please add a short description of the applicability of the indicated method to another area of research (limitations, accuracy).

Response:Thanks for pointing this out, We have added the relevant content in the section, as shown below: (line:662-685)

5 Discussion

The particular geographical environment of the Qinghai-Tibet Plateau leads to the glacial lake being frozen in most cases. In contrast, most of the existing methods aim to extract non-frozen lake boundaries and the spectral features of frozen lakes are similar to those of glaciers[65]. Therefore, this paper uses the classification extraction superposition method to extract the glacial lake boundaries. According to the geographical conditions and glacial lake distribution in Nyalam County, the classification method of classification extraction superposition of glacial lake information is proposed under step-by-step iterative remote sensing information extraction. This method overcomes the interference of common unfavourable factors such as glaciers and mountain shadows on glacial lake information extraction. It effectively improves glacial lake information extraction detection speed and quality. The glacial lake is classified as frozen and unfrozen, and the NDWI method is used to extract the unfrozen glacial lake boundary. The frozen glacial lake boundary is extracted using the difference between the frozen surface in the red band and NIR, which are more significant than the appropriate threshold and the red band is more effective than a specific value. Finally, the unfrozen and frozen glacial lake boundaries are superimposed to obtain the complete glacial lake boundary. Some studies used object-oriented image processing methods to extract glacial lakes. This paper proposes an innovative analysis method for classification and stacking extraction, which improves the extraction accuracy (Kappa coefficient reaches 90%). This method can perform spatiotemporal analysis on a cloud platform with high speed and efficiency. It can overcome the shortcomings of incomplete relevant information, long processing time, and repetitive tasks and improve the extraction efficiency of long-sequence glacial lakes. At the same time, the classification stack extraction method proposed in this study has the potential to be extended to other debris-covered glacial environment observations. However, since this paper's extraction method for unfrozen glacial lakes is based on the traditional NDWI method, the extraction of unfrozen glacial lakes is limited mainly by this method, especially in areas with severe snowfall in winter. When snow accumulates on the glacial lake, it is difficult to distinguish the specific boundary of the glacial lake, and the threshold needs to be adjusted according to the regional characteristics. The method is subsequently improved to obtain a better way of extracting the boundary of glacial lakes.

Previous studies reported that glacial lakes began to expand with climate warming but did not provide specific changes and causes of lake expansion[66]. In this paper, the characteristics of climate change and glacial lakes' temporal and spatial changes are coupled to analyze. When analyzing the driving factors of glacial lake area changes, it is considered that there may be correlations between influencing factors and the interaction of influencing factors on dependent variables. The geographic detector detected the climatic factors affecting the area change of the glacial lake, and then the multiple linear regression analysis was carried out. The response mechanism of glacial lakes in the Nyalam area to climate change and the differences in response mechanisms between different regions are preliminarily revealed. Additionally, a combined analysis of glacial lake area change, water volume change, and climate change was conducted to predict future changes. Based on the monitoring of the water level, the flow of the glacial lake in Nyalam County and the analysis of the corresponding climate, it was found that the lake showed a significant expansion between 1999 and 2010. Because the change in glacial lake area depends on the balance of water revenue and expenditure[67], it is closely related to climate change[68]. The warm season temperature increase will replenish glacier and seasonal snow meltwater to the glacial lake and promote the evaporation dissipation of the lake water when the evaporation of the glacial lake is less than the recharge of the glacial lake, making the glacial lake area decrease. The increase in cold season temperature may cause the ice and snow temperature to increase to some extent, accelerating the melting rate of ice and snow during the warm season. The increase in precipitation also means that the recharge of the glacial lake directly or indirectly by snowmelt through slope runoff increases. Among them, the water level of the glacial lake from 2019 to 2020 is highly similar to the temperature trend, which means that the temperature change is likely to cause the melting of the glaciers. This melting is the main reason for the region's expansion of the glacial lake. Because of the strong correlation between temperature and water volume and the multi-peaks distribution of particle size in glacial lake sediment. This paper infers that the temperature increase is not only the leading cause of glacial melting but also the thawing of frozen soil. This provides a preliminary research conclusion with specific reference significance for studying glacial lakes in the area. In terms of glacial lakes' spatiotemporal characteristics and their response to climate, this study mainly used mathematical analysis methods such as linear fitting, numerical comparison, and multivariate joint analysis for analysis and comparison. Methods are complex, and it is also necessary to introduce confidential data to reveal glacial lakes' temporal and spatial evolution from a deeper dimension.

Reviewer 2 Report

The authors are attempting a worthy project understanding glacial lake changes and controlling factors in Nyalam Region, China. There is a consistent lack of presented validation material for lake area identification and discharge values.  The number of lakes detected does not appear to be a realistic variation. Given the authors are focused on the hazards from the larger lakes, I would encourage them to simplify their approach and focus just on larger lakes.  Further unless they can validate the discharge model skip this part of the study. The climate factors could end up having higher correlations when related to just the 5-8 largest lakes that are in contact with retreating glaciers.

I began suggesting changes to sentences to clarify.  This is needed in most sentences.  I am happy to do this for the entire paper, when it is overall closer to being a worthwhile publication.

47: “..temperature warming rate..”

51: “..as glaciers play the role of "Asia's water tower,"

52:  “At the same time, During the x-x period most of the glaciers in….”

62:  To effectively map glacier lakes across a mountain region, optical remote sensing imagery….,

 75: “often” instead of  “always”

 76: Do you mean to say that because the most common lake size is small or below 0.1 km2?

 78: The lakes are not heavily covered by debris, do you mean that they occur in regions of extensive debris and moraine?

 96:  There are questions about the impact and forecasting of GLOF, but the number of studies is extensive not limited.

 113: reserves of what?

114: List a few key rivers.

123: At what time are there 119 glaciers and 366 lakes?

125: Specify what vicinity.

135: this is also the season with low cloud cover and lake ice cover.

178 and 183: for a, b and c what is the range around these values that are used?

194:How many lakes was validation run on? What was the result?

 232: Speed is relative, what is the time required here?

255: Any validation for this section?

 314: The number of lakes varies considerably both declining 2000-2010  and increasing  2010-2020, this appears to be unusually noisy.  How robust are these numbers?

 360: Reference? How were ice avalanches disasters quantified?           

 363: What are ice avalanche conclusions based upon?

436: The area change of glacier lakes is not linear, how well can you utilize linear trends to match a non-linear response of glacial lakes?

 488: Figure 10 indicates mean wind speed and winter temperature have the poorest correlation, not sure how that fits with statement here.

Author Response

Comments and Suggestions for Authors

Comment 1: 47: “..temperature warming rate..”

Response: Thank you for your suggestions. We have rewritten the sentence as follows:(line:50-54)

Glacial lakes, which are formed by the convergence of meltwater, are essential freshwater resources[1], and important causes of natural disasters in the Tibetan region[2]. In recent years, the rising rate of temperature in the Qinghai-Tibet Plateau has exceeded that in other areas of the same latitude in China[3; 4], leading to significant changes in the number and size of glacial lakes [5; 6].

Comment 2: 51: “..as glaciers play the role of "Asia's water tower,"

Response: Thanks for pointing this out. We have rewritten the sentence as follows:(line:54-56)

Changes in water resources and the water cycle are substantial in a warming climate[7], as glaciers play the role of "Asia's water tower," supplying water to more than 1.4 billion people[8; 9].

Comment 3: 52: “At the same time, During the x-x period most of the glaciers in….”

Response: Thanks for pointing this out, we again consulted the literature to clarify the point in time when glaciers began to retreat significantly in the Himalayan region,We have rewritten the sentence as follows:(line:56-62)

At the same time, most of the glaciers in southeastern Tibet began to shrink and melt in the mid-19th century[10], Especially after 2000, the glaciers and materials in the eastern and southern parts of the Qinghai-Tibet Plateau lost a lot and were basically in a state of accelerated loss[11], which led to the expansion of glacial lakes and the increasing frequency of their movements[12; 13].These movements resulted with an increasing number of natural disasters and the risk of natural disasters driven by glacial lake outbursts[14; 15].

Comment 4: 62: To effectively map glacier lakes across a mountain region, optical remote sensing imagery….,

Response: We appreciate the comments and thank you pointing this out. We have carefully read the text and revised this question. The revised sentence is as follows:(line:69-74)

To effectively map glacier lakes across a mountain region, optical remote sensing imagery has been used to estimate ice phenology in many lakes and rivers around the globe compared to in situ measurements[22]. The Moerate Resolustion Imageing Specytroratiometer (MODIS) was regarded as the most suitable instrument for the estimation of lake ice phenology given its outstanding ability of providing daily global images [23].

Comment 5: 75: “often” instead of “always”

Response: Thanks for pointing this out. We have rewritten the sentence as follows:(line:83-86)

More importantly, cloud-free images often suffer from huge shadows caused by highly rugged terrain, and these conditions lead to significant uncertainties in the optical images and further affect the contouring accuracy[26; 27].

Comment 6: 76: Do you mean to say that because the most common lake size is small or below 0.1 km2?

Response: Thanks for your comments,As shown in the figure, most of the glacial lakes in the study area are small or less than 0.1 km2 in size. Due to the small size and mostly glacial distribution along mountain glaciers in southwestern Tibet, most glacial lakes in southwestern Tibet are heavily covered by debris, consisting mainly of rocky debris and surface moraines. Cloud-free images are therefore often affected by large shadows caused by highly rugged terrain, and these conditions lead to significant uncertainties in the optical images and further affect the accuracy of the contours.

Comment 7: 78: The lakes are not heavily covered by debris, do you mean that they occur in regions of extensive debris and moraine?

Response: Thank you for pointing this out, we have rewritten the sentence as follows:(line:86-88)

The part of glacial lakes in southwestern Tibet appears in extensive debris and moraine areas[28], and the surrounding area of the lakeshore is covered by debris, mainly composed of rock fragments and surface moraine [19; 29].

Comment 8: 96: There are questions about the impact and forecasting of GLOF, but the number of studies is extensive not limited.

Response: We appreciate the comments and thank you for pointing this out. We have carefully read the text and revised this question. The revised sentence is as follows:(line:100-108)

Considering the potential impact of glacial lakes on the local population in the southwestern Tibetan Plateau[36], it is significant to study the changes in glacial lakes over the past 30 years[37]. Several GLOF studies have found that glacial retreat was changing water flow patterns, affecting the incidence of glacial lake outburst floods and increasing the risk of flooding and water scarcity associated with future climate change[38]. However, to accurately predict the changes of glacial lakes, it is necessary to extract their distribution characteristics and laws of glacial lakes and understand the dynamic response of climate change to hydrological processes[39; 40], Further outburst hazard studies on glacial lakes are currently lacking[41].

Comment 9: 113: reserves of what?

Response: Thank you for pointing this out. According to statistics from the China Glacier Catalogue, there are 36,793 modern glaciers in China on the Qinghai-Tibet Plateau, accounting for 79.5% of the total number of glaciers in China. The glacier area is about 49,873 square kilometers, accounting for 84% of the total glacier area in China. The ice reserves are about 4,561 cubic kilometers, accounting for 81.6% of China's ice reserves. In the Himalayas on the southern margin of the plateau, the Karakoram Mountains in the west, and the western section of the Kunlun Mountains in the north, the glaciers are most concentrated, with a large number of glaciers and a large scale of action. The revised sentence is as follows:(line:124-140)

The study area is located in the Qinghai-Tibet Plateau in southwestern China. The glaciers in this area account for about 81.6% of China's total reserves, and it is one of the areas with more ice avalanches in China[42]. These glaciers are the source of many rivers[8], such as PumQu, Boqu, Tamba Kosi and Lapchekhun Khola, Still, they are also essential in glacial lake disasters, including many steep mountains and fault zones, which are more likely to cause ice avalanches[43]. This paper selects the upper reaches of Poqu, Nyalam County, Shigatse Region, and Tibet Autonomous Region to understand further the various characteristics of glacial lakes in this region. The corridinate of study area is 86°00′00″E~86°15′00″E and 28°00′00″N~28°20′00″N. Nyalam County is located in the southwestern part of the Tibet Autonomous Region, near the edge of the Himalayan Mountains and close to the Lhari Gangri Range, with 2100 km2 surface area. In 2020, 119 glaciers and 366 glacial lakes were recorded in Nyalam County. These glaciers are mainly located in the northeast and south-central part of the county, with 98.63km2[44]. Glacial lakes are mainly concentrated in Nyalam County, the northeastern part of Nyalam County, and the junction of Yala township glacial lake distribution area is relatively large. Most of them are moraine lake or glacier blockage lakes, such as PacuCo, Chama QudanCo, DareCo, and etc. The total area of the glacial lake is 4.69km2.

Comment 10: 114: List a few key rivers.

Response: Thanks for your comments, We have added a few major rivers as examples:(line:126-129)

These glaciers are the source of many rivers[8], such as PumQu, Boqu, Tamba Kosi and Lapchekhun Khola, Still, they are also essential in glacial lake disasters, including many steep mountains and fault zones, which are more likely to cause ice avalanches[43].

Comment 11: 123: At what time are there 119 glaciers and 366 lakes?

Response: Thank you for your suggestions. We have rewritten the sentence as follows:(line:135-137)

In 2020, 119 glaciers and 366 glacial lakes were recorded in Nyalam County. These glaciers are mainly located in the northeast and south-central part of the county, with 98.63 km2[43].

Comment 12: 125: Specify what vicinity.

Response: Thank you for your suggestions. We have rewritten the sentence as follows:(line:137-140)

Glacial lakes are mainly concentrated in Nyalam County, the northeastern part of Nyalam County, and the junction of Yala township glacial lake distribution area is relatively large. Most of them are moraine lake or glacier blockage lakes, such as PacuCo, Chama QudanCo, DareCo, and etc.

Comment 13: 135: this is also the season with low cloud cover and lake ice cover.

Response: Thank you for your suggestions. We have rewritten the sentence as follows: (line:151-154)

The Landsat multispectral images from 1990 to 2020 were used to extract the number and area of glacial lakes. The remote sensing images data of the autumn and winter from September to November are selected when the glacial lake morphology is relatively stable. This is also the season with low cloud cover and lake ice cover.

Comment 14: 178 and 183: for a, b and c what is the range around these values that are used?

Response: We appreciate the comments and thank you for pointing this out. We have carefully read the text and revised this question. The range of thresholds a, b, and c used in this paper is Nyalam County in southwestern Tibet. If extracting glacial lakes in other areas of Tibet, it is necessary to adjust the threshold to increase the extraction accuracy.

Comment 15: 194: How many lakes was validation run on? What was the result?

Response: Thank you for your pointing out. We list 63 lakes in the appendix.

Table 6. Evaluation of the accuracy of the glacial lake in different states

Status

Misclassification error (%)

Omission error (%)

Kappa coefficient

Half Icing

0.67

1.40

0.86

Total Icing

0.55

1.08

0.90

Unfrozen

0.23

0.86

0.92

The accuracy evaluation of the confusion matrix is mainly used to compare the classification results with the actual measured values. The accuracy of the classification results can be displayed in a confusion matrix. Each column represents the predicted value, that is, the glacial lake extracted in this paper, and each row represents the actual value, that is, the glacial lake in the HMA Glacial Lake Inventory (Hi-MAG) database. In the table 1, 1 refers to the grid pixel value of the non-glacial lake type in the study area, and 2 refers to the grid pixel value of the glacial lake type in the study area. By comparing the position and classification of each measured pixel with the corresponding position and classification results in the classified image, the Kappa coefficient finally obtained by calculation is 0.90.

Data set:Annual 30-meter Dataset for Glacial Lakes in High Mountain Asia from 2008 to 2017 (DOI:10.5194/essd-2020-57)

The Confusion Matrix for the extraction accuracy of Glacial Lake is as follows:

Table 1. The Confusion Matrix for the extraction accuracy of Glacial Lake

Forecast Grid

Reference Grid

1

2

Sum

1

62615

141

62756

2

86

618

704

Sum

62701

759

63360

Producer Accuracy

99.86%

81.42%

User Accuracy

99.78%

87.78%

Kappa Coefficient

0.898260375

Comment 16: 232: Speed is relative, what is the time required here?

Response: Thanks for pointing this out. This paper uses SPSS software to perform multiple regression analyses on the sorted data, and its running time is speedy. The data source is the data of 30 years of climatic factors and the change of glacial lake area in the study area, and the software can be completed in 6 seconds.

Comment 17: 255: Any validation for this section?

Response: Thanks for pointing this out, I believe it is because I did not express it clearly enough in the manuscript. The surface of GangxiCo glacier lake is olive-shaped, and the lake area runs from west to east. The measured maximum water depth is in the middle area on the west side of the lake area, and the real-time maximum water depth is 49.1m. Combined with remote sensing images and downstream longitudinal section measurement results, it is estimated that the slope of the river channel is S=0.048, and the channel sediments are mainly coarse sand and gravel formed by the erosion of water flow after the weathering of granite gneiss. Referring to the "Hydraulic Calculation Manual" (Second Edition), select the roughness ratio of n=0.04. Finally, combined with the "Tibet Autonomous Region Shigatse Glacial Lake Surveying and Mapping Project," the water level-flow formula of the outlet section and the elevation obtained by the Manning formula is the relationship formula of the channel in the lower reaches of GangxiCo. We have rewritten the sentence as follows:(line:301-308)

The channel sediments are mainly coarse sand and gravel formed by water scouring after the weathering of granite gneiss[47], referring to the "Hydraulic Calculation Manual" (Second Edition), and the roughness ratio of n=0.04 is comprehensively selected according to the cross-sectional shape of the river bed and the structure of the beach[48].Finally, combined with the "Tibet Autonomous Region Shigatse Glacial Lake Surveying and Mapping Project," the water level-flow formula of the outlet section and the elevation obtained by the Manning formula is the relationship formula of the channel in the lower reaches of GangxiCo.

Comment 18: 314: The number of lakes varies considerably both declining 2000-2010 and increasing 2010-2020, this appears to be unusually noisy. How robust are these numbers?

Response: We appreciate the comments and thank you for pointing this out. We have rewritten the sentence as follows: (line:358-364)

Between 2000 and 2010, the number of lakes decreased, primarily due to the disappearance of small glacial lakes near steep mountains and the merging of broken glacial lakes due to the increase in glacial meltwater. Therefore, while the number decreased, the area continued to increase[49]. After 2010, smaller glaciers were more sensitive to climate change because smaller glaciers reduced their area at a higher rate than larger glaciers[50]. According to the current situation, glaciers will further decrease, and the area of glacial lakes will continue to increase[51].

Comment 19: 360: Reference? How were ice avalanches disasters quantified?

Response: We appreciate the comments and thank you for pointing this out. The ice avalanches data comes from the statistical dataset of significant geological hazards in the Himalayas. National Qinghai-Tibet Plateau Scientific Data Center. We have rewritten the sentence as follows: (line:406-412)

The loss of glacier mass balance can quantify the information of glacial avalanche hazards[53]. The primary trend in glaciers in the Himalayas and central and southern Tibetan Plateau is overall mass loss[54]. Based on historical glacial lake outburst data and visual interpretation[55; 56], the distribution characteristics and types of glacial lake outbursts in Nyalam County were extracted and analyzed. The southern mountain range's edge was where Nyalam County's ice avalanche disasters were mainly concentrated.

Comment 20: 363: What are ice avalanche conclusions based upon?

Response: Many glaciers in this area have created conditions for ice avalanches. Generally speaking, glacial lakes are glacial moraine lakes that end close to supply glaciers and are more prone to outbursts. Ice avalanches occur when potentially dangerous moraine-dammed lakes develop behind unstable ice-cored moraines. It is also likely to have been released directly from glaciers or erupted from glacial and moraine lakes. This study found that the ice avalanche disasters in the study area were mainly concentrated in the edge areas of the southern mountain range, and many glaciers in this area created conditions for the occurrence of glacial avalanche disasters, indicating that the glacial lakes that are more prone to ice avalanches are moraine lakes with ends close to supply glaciers. We have rewritten the sentence as follows:(line:406-414)

The loss of glacier mass balance can quantify the information of glacial avalanche hazards[53]. The primary trend in glaciers in the Himalayas and central and southern Tibetan Plateau is overall mass loss[54]. Based on historical glacial lake outburst data and visual interpretation[55; 56], the distribution characteristics and types of glacial lake outbursts in Nyalam County were extracted and analyzed. The southern mountain range's edge was where Nyalam County's ice avalanche disasters were mainly concentrated. Many glaciers in the region create conditions for glacial avalanche disasters[57]. Glacial lakes that are more prone to outbursts are glacial moraine lakes that end close to supply glaciers[58].

Comment 21: 436: The area change of glacier lakes is not linear, how well can you utilize linear trends to match a non-linear response of glacial lakes?

Response: We appreciate the comments and thank you for pointing this out. This paper uses a linear regression model to analyze the changes in climate factors over time. The linear change trend of each climate influencing factor in the past three decades can be observed through the chart, and the 1990-2020 glacial lake area data can be tested using Mann-Kendall. The standardized test statistic Z is defined by calculation. When the absolute value of Z was greater than 1.65, 1.96, and 2.58, the trend passed the significance test with 90%, 95%, and 99% reliability, respectively. After testing, Z=2.73> 2.58, which indicates that there is an apparent trend change in the data. And p = 0.007 < 0.05, indicating that the time series data has a trend. In addition, the change in the Sen slope indicates that the glacial lake area is gradually increasing (Sen's slope=4.16), indicating that the area of the glacial lake in the study area has increased linearly and significantly over the past three decades. Considering Sen's slope, the method of multiple linear regression analysis was selected to explore the relationship between each climatic factor and the change in the glacial lake area.

Comment 22: 488: Figure 10 indicates mean wind speed and winter temperature have the poorest correlation, not sure how that fits with statement here.

Response: Thank you for your suggestions. This paper uses geographic detectors and multiple linear regression analysis models and analyses the significant relationship of various climatic factors to the area change of glacial lakes. We have rewritten the sentence as follows: (line:547-549)

Figure 10. Correlation of climate factors

Figure 10 shows that the significance ranking of the variation of a glacial lake in the Nyalam County area is mean relative humidity > mean temperature > sunshine hours > precipitation> mean wind speed. Overall, the average annual climate is mainly influenced by the winter temperature. The average annual temperature from 1990 to 2020 shows a slowly increasing trend with a climate tendency rate of 0.34℃(10 years). The annual precipitation shows a weak (insignificant) increasing trend, and the climate trend rate is 5.64mm/(10 years). The annual temperature is positively correlated with the change in the glacial lake area, and the annual precipitation is negatively correlated with the difference in the glacial lake area (r=-0.23). However, the glacial lake area change is highly correlated with summer precipitation (r=-0.57), indicating that the area varies with the seasons. Precipitation is also one of the essential reasons for the change of glacial lakes. Since the humidity is highly correlated with the area change (r=0.56), the variance of the average relative humidity and the sunshine change obtained by the univariate linear model is slight, and the annual difference is not apparent. Therefore, the changes in glacial lakes in Nyalam County are mainly affected by the changes in temperature and precipitation. From 1990 to 2020, the temperature and precipitation in Nyalam County showed an increasing trend. Temperature and precipitation are meteorological factors with high correlation. The continuous increase in temperature is the dominant factor in the melting of glaciers, and the melting of glaciers has caused an increase in the area of glacial lakes.

Reviewer 3 Report

The manuscript “Characterization of Long-time series variation of glacial lakes in Southwestern Tibet” by Ge et al. surveyed the glacial lake variation and driven factors by Landsat Images and meteorological data. However, there are carefulless in writing, I suggest it should be major revision.

Special comments

Reference:

W. Wang, Y. Gao, P. Iribarren Anacona et al., Integrated hazard assessment of Cirenmaco glacial lake in Zhangzangbo valley, Central Himalayas[J], Geomorphology, 2018, 306: 292-305. https://doi.org/10.1016/j.geomorph.2015.08.013

Line 37 431,042*103->3.7*108

Line 11 section 2.1 study Area  This section is not fitable in “Materials and Methods”

Line 140 section 3 Methods? What ‘s difference with section 2?

Figure 1 Glacial lake area change in study region should be omitted.

Table 1 It should be informed the time of RS imageries.

Figure 2 Frozen lake? In the result section, there is not Frozen lake result. I would suggest it should be omitted.

Table 3 there are repeated information here. Please check it carefully.

Figure 8 the text in Legend is too small, and can’t read it.

Table 6 please give the station information, such as longitude, latitude et al.

Figure 10 why did authors select those parameters(annual T, summer T, winter T and so on…), Is the correlation analysis being performed?

Line 487 above table, which table? Influence area?, I can not see the mean relative humidity > mean wind speed?

Figure 12  where is contour line elevation in legend?

Author Response

Comments and Suggestions for Authors

Comment 1: Special comments, Reference: W. Wang, Y. Gao, P. Iribarren Anacona et al., Integrated hazard assessment of Cirenmaco glacial lake in Zhangzangbo valley, Central Himalayas [J], Geomorphology, 2018 , 306 : 292-305. https://doi.org/10.1016/j.geomorph.2015.08.013

Response: Thank you for your references. we has added the reference in the manuscript. I will pay attention to this in later study.This study suggests that the potential triggers for GLOF are mass movement and dead ice melting, which aligns with our view that moraine lakes that end close to supply glaciers are more prone to ice avalanches.

Comment 2: Line 37 431,042*103->3.7*108

Response: Thank you for your suggestion. We have corrected the presentation of the numbers so that it looks more concise and clear. The revised sentence is as follows:

(line:42-45)

It is concluded that GangxiCo endures a maximum water flow of 4.3×108m3, and the glacial lake is in a stable changing stage. This conclusion is consistent with the realistic investigation and can be used to provide scientific guidance for predicting glacial lake outbursts in Southwest Tibet in the future.

Comment 3: Line 110 section 2.1 Study Area This section is not fitable in “Materials and Methods”

Response: Thanks for pointing this out. We have changed the name of section 2 to "Study area and data". We will pay attention to this in later study. We have rewritten the name of this section as follows:(line:122)

2.Study area and data

2.1. Study Area

The study area is located in the Qinghai-Tibet Plateau in southwestern China. The glaciers in this area account for about 81.6% of China's total reserves, and it is one of the areas with more ice avalanches in China[42]. These glaciers are the source of many rivers[8], such as PumQu, Boqu, Tamba Kosi and Lapchekhun Khola, Still, they are also essential in glacial lake disasters, including many steep mountains and fault zones, which are more likely to cause ice avalanches[43]. This paper selects the upper reaches of Poqu, Nyalam County, Shigatse Region, and Tibet Autonomous Region to understand further the various characteristics of glacial lakes in this region. The corridinate of study area is 86°00′00″E~86°15′00″E and 28°00′00″N~28°20′00″N. Nyalam County is located in the southwestern part of the Tibet Autonomous Region, near the edge of the Himalayan Mountains and close to the Lhari Gangri Range, with 2100 km2 surface area. In 2020, 119 glaciers and 366 glacial lakes were recorded in Nyalam County. These glaciers are mainly located in the northeast and south-central part of the county, with 98.63 km2[44]. Glacial lakes are mainly concentrated in Nyalam County, the northeastern part of Nyalam County, and the junction of Yala township glacial lake distribution area is relatively large. Most of them are moraine lake or glacier blockage lakes, such as PacuCo, Chama QudanCo, DareCo, and etc. The total area of the glacial lake is 4.69km2.

The geomorphology of the study area presents an alternating pattern of alpine valleys and plateau lake basins. Among them, large glacial lakes are formed in a concentrated distribution near the alpine glaciers, and small glacial trough lakes and ice lakes are scatterly distributed in the valleys. The region mainly has a sub-arctic semi-arid climate and a humid subtropical climate. The dry season has a distinct rainy season. The rainy season is in June and August while the snowfall lasts for six months. The huge rainfall is likely to cause the glacial lake outburst.

Comment 4: Line 140 section 3 Methods? What ‘s difference with section 2?

Response: Thanks for pointing this out, we have corrected the name of the section in Line 110 section 2 to "Study area and data". Section 2 contains the study area and data sources and preprocessing, line 140 Section 3 methods are the methods of this paper.

Comment 5: Figure 1 Glacial lake area change in study region should be omitted.

Response: Thanks for pointing this out, we will carefully check the Figure 1 and adjusted the statement of the sentence.

Figure 1.Study area and distribution of glacial lake area change

Comment 6: Table 1 It should be informed the time of RS imageries.

Response: Thank you for your suggestions. We have added the Table1(a) about the RS imageries of data sources.

Table 1(a). Remote sensing data of the study area

Sensor

Imaging time

Orbital parameters

Resolution(m)

Landsat4/5 TM

1990.9.15

138/039

30/120

Landsat4/5 TM

1995.10.13

138/039

30/120

Landsat7 ETM+

2000.11.4

138/039

15/30

Landsat5 TM

2005.11.12

138/039

30/120

Landsat8 OLI

2010.11.16

138/039

15/30

Landsat8 OLI

2015.11.24

138/039

15/30

Landsat9 OLI

2020.10.9

138/039

15/30

GF-1

2018.10.10

8/16

GF-1

2019.10.5

8/16

Table 1(b). Data of the study area

Data classification

Name

Resolution

Data source

Remote sensing imagery data

Landsat

30m

USGS

(https://glovis.usgs.gov/)

GF

16m

Geospatial Data Cloud(http://www.gscloud.cn/)

SRTM

30m

USGS

(https://glovis.usgs.gov/)

Meteorological data

Meteorological station

Forms for Report

Meteorological Data Center of China Meteorological Administration (http://cdc.cma.gov.cn/home.do)

Zone

Administrative boundary vector

Vector Data

Resource and Environment Science and Data Center(https://www.resdc.cn/)

Comment 7: Figure 2 Frozen lake? In the result section, there is not Frozen lake result. I would suggest it should be omitted.

Response: Thanks for pointing this out, I believe it is because I did not express it clearly enough in the manuscript.

Figure 2. Flow chart of the methodology

Comment 8: Table 3 there are repeated information here. Please check it carefully.

Response: Thanks for pointing this out, we have removed the repetitive parts and will pay attention to this in future studies.

Table 3. Change of glacial lakes for different elevations

Altitude(m)

Current (2020) total area (km2)

Moraine-dammed lake

U-type valley lake

Cirque lake

Glacial erosion lake

Lateral moraine lake

Total area of increase (km2)

Number of lakes showing areal increases

Number of lakes showing no increase in area

< 5000

1.44

6

2

1

1

0.314

3

7

5000–5100

7.91

3

1

1

1

3.017

5

1

5100–5200

3.45

5

3

2

0.106

7

5

5200–5300

7.51

12

1

2.384

10

3

5300–5400

14.31

14

1

1

2.263

14

2

>5400

1.109

8

0.006

2

6

Comment 9: Figure 8 the text in Legend is too small, and can’t read it.

Response: Thanks for pointing this out, we have enlarged the image size to make it easier to see the legend text, as shown below:

Figure 8(a). Interpretation map of GangxiCo glacial lake and surrounding in 2018

Figure 8(b). Interpretation map of GangxiCo glacial lake and surrounding in 2019

Comment 10: Table 6 please give the station information, such as longitude, latitude et al.

Response: Thanks for pointing this out, we have removed the repetitive parts and will pay attention to this in future studies. The revised sentence is as follows:(line:460-462)

The expansion of the glacial lake in the Nyalam is closely related to external climate change. This paper analyzes the glacial lakes' evolution characteristics are statistically sorted by periods. The average statistics of climate factors are calculated in each decade as a period, and the climate change rate in different periods is calculated by linear fitting based on the annual data. By sorting out and calculating the changes in the glacial lake area and the changes in the corresponding climatic factors, it is found that the GangxiCo in Nyalam County has expanded by a total of 1.88 km. Among them, the area of glacial lakes expanded by 0.699 km2, 1.051 km2, and 0.13 km2 respectively in 1990-2000, 2000-2010, and 2010-2020. The statistics of the growth area and the average value of meteorological factors in the three periods are shown in Table 6. The nearest observed meteorological station in the Nyalam area to the glacial lake: Nyalam station was selected(85°58′E, 28°11′N).

Table 6. Thirty-year annual change in Nyalam County

1990-2000

2000-2010

2010-2020

1990-2020

Glacial lake growth area (km²)

0.699

1.051

0.13

1.88

Annual temperature (℃)

3.69

4.312

4.158

4.042

Summer temperature (℃)

10.2919

10.6067

10.4191

10.441

Winter temperature (℃)

-2.897

-1.673

-1.987

-2.2

Annual precipitation (mm)

551.76

613.08

632.93

595.51

Summer precipitation (mm)

203.63

198.86

220.95

206.36

Precipitation in winter (mm)

99.72

102.79

141.1

111.6

Sunshine hours (h)

2515.49

2477.95

2507.76

2499.58

Average wind speed (m/s)

4.686

4.063

4.069

4.29

Average relative humidity (%)

67.39

67.71

62.9

66.34

Glacial lake growth area (km²)

0.699

1.051

0.13

1.88

Comment 11: Figure 10 why did authors select those parameters (annual T, summer T, winter T and so on…), Is the correlation analysis being performed?

Response: Thank you for pointing this out. Sorry for the unclear statement. We have rewritten it in the revised manuscript.(line:489-501)

Changes of Glacial lakes are closely related to global climate change. Glaciers recede when the weather warms up, supplying high lakes with meltwater as the lakes grow in size.[59]. Seepage lubricates the glacier bottom during severe melting, which may lead to frontal collapse if the glacier extends into the lake, thus triggering lake eruption. The expansion of TP glacial lakes can be explained by the trend of increased precipitation and accelerated melting of glaciers associated with rising temperatures[60]. Many studies have provided substantial evidence for lake and glacier dynamics on the TP under climate fluctuations[61]. Some researchers suggest that increased regional precipitation between 1990 and 2010 led to a significant expansion of enclosed lakes on the TP[62]. At the same time, others argue that the expansion of lakes fed by glaciers is primarily caused by rising temperatures that accelerate glacier shrinkage[52; 63]. Based on the above research experience, this paper selects these climate influencing factors for correlation analysis.This paper uses data from 1990 to 2020 from meteorological stations in Nyalam County based on five climate impact factors: annual average precipitation, annual average temperature, annual average sunshine hours, annual average relative humidity, and annual average wind speed.

The correlation regression analysis procedure is as follows.:

A tolerance less than 0.1 indicates severe multicollinearity in the covariance statistics, and a variance expansion factor VIF more significant than 10 indicates severe collinearity. When the VIF value is less than 10, we believe the data conform to the multiple linear analysis, and there is no multicollinearity among the independent variables. It can be seen from the table that the VIFs of sunshine hours, average temperature, average wind speed, average relative humidity, and precipitation are all less than 10, and their tolerances are all close to 0.8. That is, there is no multicollinearity problem in this model. Analysis of variance (ANOVA) is a significant test invented by R.A. Fisher for the difference in the mean of more than one sample. This paper uses ANOVA to judge whether the regression equation is essential. The corresponding p-value in the ANOVA model, which is the sig in the table, is significant and indicates the size of the difference between the control and the experimental group. In the table, p=0.011<0.05 indicates that the null hypothesis is supported. That is, the linear regression equation is significant. The corresponding T-test below is to judge the importance of each variable of the regression equation.

Table 8. Table of coefficients

Coefficients a

Model

Unstandardized coefficient

Standardized coefficient

t

Sig.

Covariance statistics

B

Standard Error

Trial version

Tolerance

VIF

1

(Constant)

163.126

46.453

3.512

.002

Sunshine hours

-.010

.012

-.134

-.825

.419

.926

1.080

Average temperature

-2.462

2.180

-.239

-1.130

.271

.546

1.833

Average wind speed

-6.922

3.013

-.513

-2.298

.032

.490

2.040

Average relative humidity

-.961

.256

-.660

-3.758

.001

.791

1.264

Precipitation

-.006

.009

-.122

-.674

.508

.746

1.341

a. Dependent variables: area

From the above calculation, the optimized regression equation is:

(21)

The goodness of fit R2 is 0.689, and the regression coefficients of all independent variables passed the t-test, with an excellent overall fit and no multicollinearity. The criterion of standardized residuals by regression illustrates that this study's polynomial linear simulation model meets the regression requirements better.

The test results of the regression model:

A2. Standard P-P plot of regression standardized residuals

Comment 12: Line 487 above table, which table? Influence area?, I can not see the mean relative humidity > mean wind speed?

Response: Thank you for your suggestions. We have rewritten the sentence as follows:(line:547-549)

Figure 10. Correlation of climate factors

The above Figure 10 shows that the significance ranking of the variation of glacial lake in the Nyalam County area is mean relative humidity > mean temperature > sunshine hours > precipitation> mean wind speed. 

Comment 13: Figure 12 where is contour line elevation in legend?

Figure 12. Changes before and after the glacial lake outburst

Response: Thanks for pointing this out, we have rewritten the sentence as follows:In the top left diagram, the line inside the lake is the isobath and the line outside the lake is the Contour line. We have rewritten the sentence as follows:(line:625-627)

4.3 Simulation Results of a Glacial Lake Outburst Event

Based on DEM in the simulation, the physical input parameters of HEC-GeoRAS are obtained through ArcGIS analysis. The range of calibrated Manning resistivity n in the main channel is obtained using the nominal curve values exercise, and the simulation results were exported to ArcGIS to depict the water surface in the floodplain. The maximum flow values from the observatory were used for the upstream flood input to simulate a glacial lake outbursts scenario. The water surface elevation measured is 5181.01m, the contact length between the water crossing section and the water body is L=17.19m, and the water depth is h=0.15m. Taking the above parameters into the Manning formula, the flow rate of GangxiCo is about 0.67m3/s during this measurement period. Figure 14 compares the images of the glacial lake before and after the breach. At this time, the maximum storage volume can be calculated as 4.3×108m3. The flood inundation area of HEC-RAS in the simulated river reach (402.53m) is about 42.87 m2. In the top-left diagram, the line inside the lake is the isobath and the line outside the lake is the Contour line. 

Round 2

Reviewer 2 Report

The authors provide poor context and analysis vs other papers I have reviewed and read on lakes in the area see Zhang et al (2018) for example.  The information in Table 4 is simply not accurate for moraine ridge elevation or margin change. At present this paper does not advance our knowledge of the region and contains significant errors in assessment.

54; “..and permafrost in the eastern and southern parts of the Qinghai Tibet Plateau were in a state of accelerated loss.”

57: What is “their”? The glaciers, landslides the lakes?

58: What “movements”?

63: “Although a few scholars have investigated glaciers and glacial lakes in Tibet” I have reviewed more than 20 papers on this topic of Tibet glacier and glacial lakes, so this is simply not true, many have investigated these important features.

102: “Further outburst hazard studies on glacial lakes are currently lacking[41].” The paper referenced is from 2014 in the eight years since many studies have in fact done just this.

120: Provide an assessment of how many ice avalanches.

141:  What is the freezing level during the summer monsoon period?

359: I do not have confidence in the decrease in glacier lake area from 1990-1995. These lakes develop slowly and progressively, your data for all lakes does not show this.  For the biggest lakes alone does it? I also see in the one 1995 image shared an unidentified lake.

364: Figure 5 caption identify the lakes we are looking at. Label the year on each image. One lake in 1995 is not outlined, that is in the 1990 and 2000 images.  This is not supported by other work such as Zhang et al (2018) who in a study including this area show persistent growth in area and to a lesser extent number of lakes. They found that Gangxico was 4.49 km2 (+245% since 1964) and Galongco was 5.46 km2 (+447% since 1964). Nie et al (2013) have a much more detailed visual of change for Galongco, which is a template for change that should be used in this figure.

385: Figure 4 only shows Gongcuo Lake, what type of lake is this?

399: Be more specific about number timing and location of ice avalanches, these are critical events for hazards.

413: This does not make sense “and the distance of the glacial lake edge from the glacial margin decreased by 170m” The 170 m change which would be retreat is not correct.

Table 4:  We cannot take seriously a 48 m change in elevation of the moraine ridge line crest.  There is no evident change in the imagery. Even 5 m would be substantial. The 170 m change in margin is also not realistic.  Where is the comparison here with other detailed studies of Gangxico and Galongco?  Zhang et al (2019)  observed that from 1974-2014 Galongco and Gangxico lakes expanded by ~500% (0.45 km2 /year) and ~107% (0.34 km/year).  As the lakes have expanded the wide moraines impounding the lakes have not experienced visible change.

ZHANG, G., BOLCH, T., ALLEN, S., LINSBAUER, A., CHEN, W., & WANG, W. (2019). Glacial lake evolution and glacier–lake interactions in the Poiqu River basin, central Himalaya, 1964–2017. Journal of Glaciology, 65(251), 347-365. doi:10.1017/jog.2019.13

The authors provide poor context and analysis vs other papers I have reviewed and read on lakes in the area see Zhang et al (2018) for example.  The information in Table 4 is simply not accurate for moraine ridge elevation or margin change. At present this paper does not advance our knowledge of the region and contains significant errors in assessment.

54; “..and permafrost in the eastern and southern parts of the Qinghai Tibet Plateau were in a state of accelerated loss.”

57: What is “their”? The glaciers, landslides the lakes?

58: What “movements”?

63: “Although a few scholars have investigated glaciers and glacial lakes in Tibet” I have reviewed more than 20 papers on this topic of Tibet glacier and glacial lakes, so this is simply not true, many have investigated these important features.

102: “Further outburst hazard studies on glacial lakes are currently lacking[41].” The paper referenced is from 2014 in the eight years since many studies have in fact done just this.

120: Provide an assessment of how many ice avalanches.

141:  What is the freezing level during the summer monsoon period?

359: I do not have confidence in the decrease in glacier lake area from 1990-1995. These lakes develop slowly and progressively, your data for all lakes does not show this.  For the biggest lakes alone does it? I also see in the one 1995 image shared an unidentified lake.

364: Figure 5 caption identify the lakes we are looking at. Label the year on each image. One lake in 1995 is not outlined, that is in the 1990 and 2000 images.  This is not supported by other work such as Zhang et al (2018) who in a study including this area show persistent growth in area and to a lesser extent number of lakes. They found that Gangxico was 4.49 km2 (+245% since 1964) and Galongco was 5.46 km2 (+447% since 1964). Nie et al (2013) have a much more detailed visual of change for Galongco, which is a template for change that should be used in this figure.

385: Figure 4 only shows Gongcuo Lake, what type of lake is this?

399: Be more specific about number timing and location of ice avalanches, these are critical events for hazards.

413: This does not make sense “and the distance of the glacial lake edge from the glacial margin decreased by 170m” The 170 m change which would be retreat is not correct.

Table 4:  We cannot take seriously a 48 m change in elevation of the moraine ridge line crest.  There is no evident change in the imagery. Even 5 m would be substantial. The 170 m change in margin is also not realistic.  Where is the comparison here with other detailed studies of Gangxico and Galongco?  Zhang et al (2019)  observed that from 1974-2014 Galongco and Gangxico lakes expanded by ~500% (0.45 km2 /year) and ~107% (0.34 km/year).  As the lakes have expanded the wide moraines impounding the lakes have not experienced visible change.

ZHANG, G., BOLCH, T., ALLEN, S., LINSBAUER, A., CHEN, W., & WANG, W. (2019). Glacial lake evolution and glacier–lake interactions in the Poiqu River basin, central Himalaya, 1964–2017. Journal of Glaciology, 65(251), 347-365. doi:10.1017/jog.2019.13

Reviewer 3 Report

The paper has undergone principal revision and improvements since the last sub mission.